# Identification and engineering of highly functional potyviral proteases in cells using co-evolutionary models

Medel B. Lim Suan Jr[1,5], Cheyenne Ziegler [2,5], Zain Syed [2], Arjun Sai Yedavalli[2], Jaimahesh Nagineni [2], Rodrigo Raposo[2], Ajay Tunikipati[2], Jaideep Kaur[1], Faruck Morcos [1,2,3,4] ✉ & P. C. Dave P. Dingal [1,2,3] ✉

Efficiency and substrate specificity of proteases in the *Potyviridae* family have not been comprehensively profiled. Here we develop a model that learns co-evolutionary features to accurately predict and experimentally validate protease performance at single amino-acid resolution. We identify and engineer several proteases that perform better than the commercially available tobacco etch virus protease. To demonstrate the resolving power of our methods, we engineer protease crosstalk to selectively trigger a synthetic cell-death program in human cells.

Proteases are commonly used tools in protein chemistry[1,2], industrial applications[3], and as composable parts of synthetic biological circuits[4–7]. Nuclear Inclusion a (NIa) proteases are cysteine proteases in the *Potyviridae* family that can recognize a unique cleavable sequence (cs) of seven amino acids[8] (Fig. 1a, b). For example, the tobacco etch virus protease (TEVp) is known to cleave its TEVcs: ENLYFQ^S (where ^ is the cleaved peptide bond) and is commercially produced for use in biochemical purification of recombinant proteins. More than 3800 NIa proteases and variants are known, but their catalytic activities and substrate specificities have yet to be determined.

We hypothesized that co-evolutionary features of protease-substrate sequences can be used to quantitatively predict and validate the function of potyviral proteases in human cells. The co-evolutionary pressure to maintain protease-substrate specificity influences which pairs possess the sequence composition to perform cleavage and therefore persist in nature. To investigate the specificity of protease-substrate interactions, we have compiled 3817 pairs of NIa protease and substrate (Supplementary Data 1) that provide a statistical setting to build the protease substrate specificity calculator (ProSSpeC). ProSSpeC leverages direct coupling analysis (DCA) to learn patterns of specificity by estimating a joint probability distribution of sequences in interacting families. DCA has been used to study

protein structure[9], complex formation[10], conformational plasticity[11] and specificity between two protein families[12,13]. It also served as inspiration for the Evoformer module of AlphaFold2 and the Pairformer module of AlphaFold3[14,15], which have also been utilized for protein ligand docking[16]. DCA can identify epistatic interactions, and we use the collection of DCA parameters learned from paired protease-substrate sequences to construct a Hamiltonian specificity ($H_{\text{spec}}$) score (Fig. 1c). Any change in sequence composition (of both protease and substrate) affects the strength of amino acid couplings and the resulting $H_{\text{spec}}$ score, providing a quantitative measure of specificity upon mutation. We reasoned that ProSSpeC can be used to predict and design protease sequences that can cleave specific substrate sequences.

Here we show how our quantitative modeling method, ProSSpeC, can resolve single amino-acid effects on protease-substrate interaction. We first establish that ProSSpeC can predict proteolytic cleavage efficiency against any natural potyviral substrate sequence. We then show that ProSSpeC can predict changes in cleavage efficiency of substrates undergoing point mutations. Last, we apply the predictive power of ProSSpeC to engineer selective protease-induced cell death at single amino-acid resolution. Together, these results show that ProSSpeC can accelerate the design of proteases for various biomedical and biotechnology applications.

[1]Department of Bioengineering, The University of Texas at Dallas, Richardson, TX, USA. [2]Department of Biological Sciences, The University of Texas at Dallas, Richardson, TX, USA. [3]Center for Systems Biology, The University of Texas at Dallas, Richardson, TX, USA. [4]Department of Physics, The University of Texas at Dallas, Richardson, TX, USA. [5]These authors contributed equally: Medel B. Lim Suan Jr, Cheyenne Ziegler. ✉e-mail: faruckm@utdallas.edu; davedingal@utdallas.edu

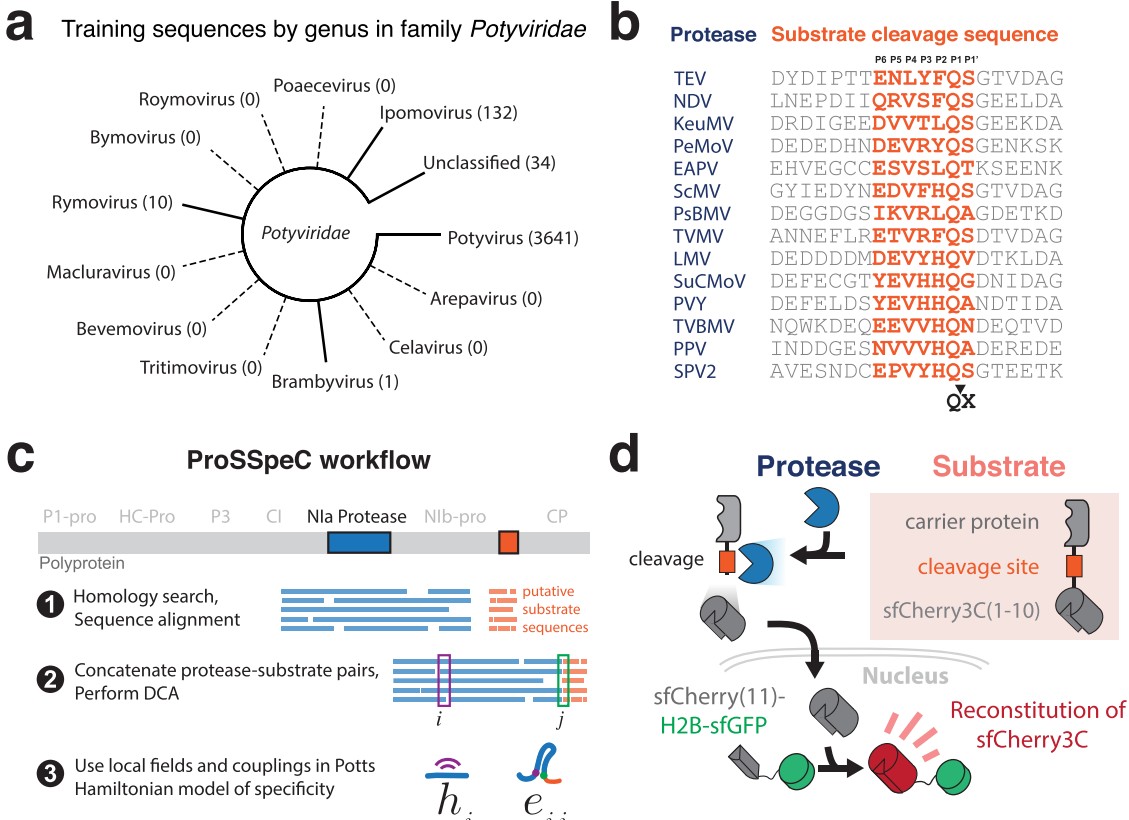

**Fig. 1 | Overview of ProSSpeC and experimental design. a** Phylogeny of the *Potyviridae* family with genera branches present in the sequence data (number of training sequences per genus); dotted lines depict branches not used in model training. **b** Substrate sequence and cleavage position for a subset of representative proteases. **c** ProSSpeC modeling workflow prior to $H_{spec}$ calculation. **d** The ability of proteases to cleave a substrate sequence was measured via reconstitution of sfCherry3C fluorescence.

## Results

### Coevolutionary modeling of native proteolytic activity

ProSSpeC is a coevolutionary model that leverages the covariation of NIa protease sequences along with their aligned substrate sequences to calculate a Hamiltonian specificity score ($H_{spec}$), where the more negative the $H_{spec}$ score is, the stronger the predicted specificity is. This specificity score is defined by the Potts Hamiltonian energy of canonical protease-substrate pairs (Eq. 1), with the Potts Hamiltonian energy of scrambled enzyme-substrate pairs (Eq. 2) removed, to provide a score of sequence attributes unique to interacting pairs and not shared due to family-likeness or general sequence similarity (Eq. 3). For both canonical and scrambled protease-substrate pairs, the Potts Hamiltonian is defined by as the sum of all parameters in a Boltzmann distribution, with couplings ($e_{ij}$) and local fields ($h_i$) are inferred using direct coupling analysis (DCA), specifically the mean-field formulation (mfDCA)[9]. A similar rationale for using the DCA Hamiltonian to characterize molecular interactions has been applied to study specificity in two-component systems[12], recognition in protein-RNA interactions[17], and to predict and engineer compatibility in hybrid transcription factors[13,18]. The ProSSpeC model applies this approach to quantify interactions of the protease and the substrate. When we were interested in only the 7-amino acid substrate motif; we masked out substrate residues outside of this window in $H_{spec}$ calculations to yield the masked $H_{spec}$ (Eq. 4). The masked $H_{spec}$ was only utilized for predictions where the designed experiment did not include the extended substrate context. The ProSSpeC model allows us to quantitatively assess the specificity of NIa proteases and their substrate motif through sequence alone, allowing us to filter and rank both natural NIa

proteases as well as mutated ones prior to experimental testing.

### Experimental validation of the ProSSpeC model

To rapidly screen for sequence-specific cleavage activity, we repurposed a fluorescent protein reconstitution assay in human cells that has been used to cleave and release protein cargoes into the nucleus[19] (Fig. 1d). We initially assayed a set of 31 protease-substrate sequence pairs (Supplementary Fig. 1), which included a few proteases with favorable (i.e., most negative) ProSSpeC $H_{spec}$ scores (Fig. 2a). A subset of these proteases exhibited cleavage-induced fluorescence that are equal to or better than that of TEVp, and they also possessed similar or more favorable $H_{spec}$ scores than TEVp (Fig. 2b). This result supports our hypothesis that ProSSpeC can infer proteolytic cleavage activity from co-evolutionary sequence features of protease-substrate families.

To further determine if ProSSpeC can estimate protease specificity against any substrate sequence, we determined the specificity of fifteen proteases against fifteen substrates in human cells. We found that favorable $H_{spec}$ scores arise when a protease is paired with its putative native substrate sequence (Fig. 3a). For example, when pairing TEVp with each of the fifteen substrates, the TEVp-TEVcs pair has the lowest $H_{spec}$ score ($H = -147$; $H$-range of $[-147, 1]$) and exhibited the highest cleavage-induced fluorescence intensity than any other TEVp-substrate pairs. Importantly, the fluorescence intensities of all 225 protease-substrate pairs are correlated with $H_{spec}$ scores (Fig. 3b and see also Supplementary Fig. 2). Also, using these pairs and to quantify how the model captures crosstalk in off-diagonal elements, we generated a receiver operating characteristic (ROC) curve with an area

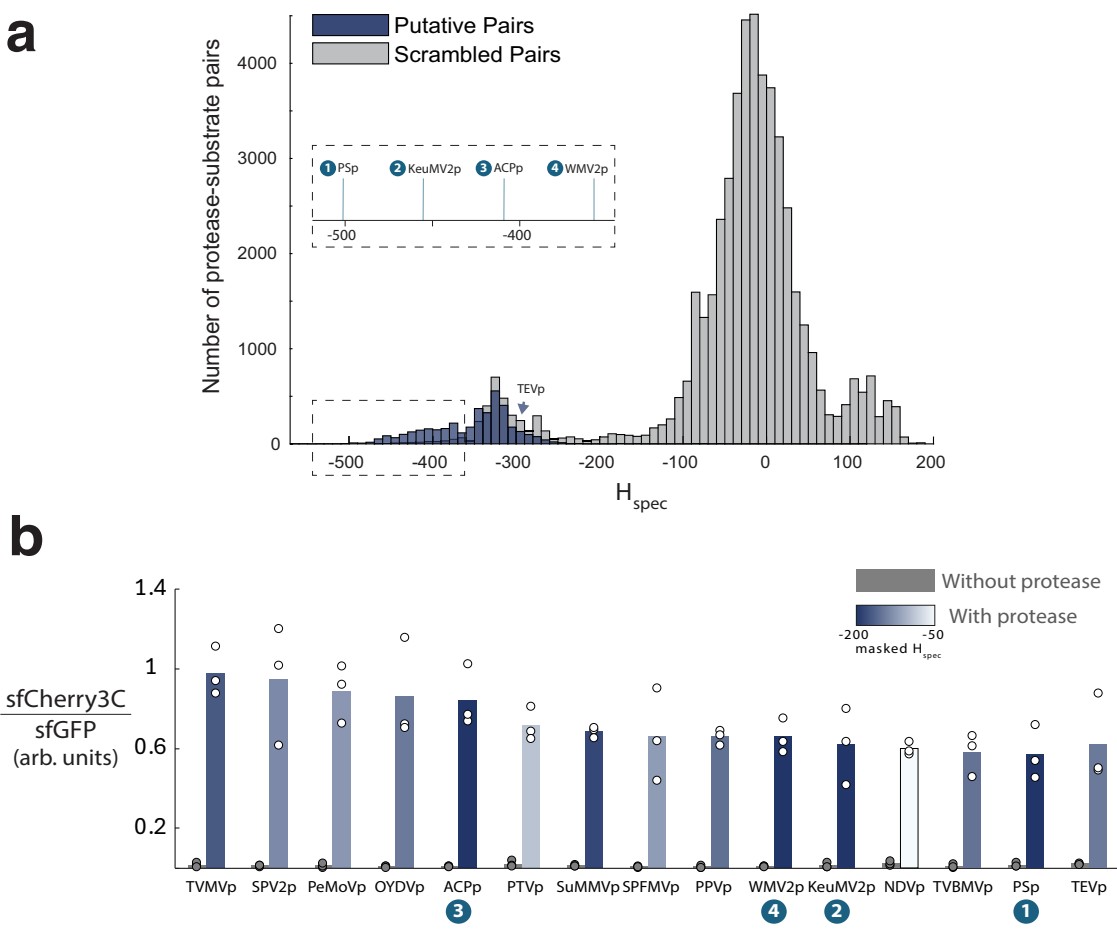

**Fig. 2 | Experimental validation of protease-substrate interactions. a** $H_{spec}$ distributions for putative cognate protease-substrate pairs (blue) and scrambled pairs (gray). More negative $H_{spec}$ scores indicate more favorable fitness, as defined by the ProSSpeC model. Inset: ProSSpeC predicted four proteases with stronger interactions with their cognate substrates than TEVp. **b** Normalized cleavage-induced fluorescence for a subset of proteases predicted to be similar or better than TEVp. Data are shown as mean values of $n = 3$ independent replicates. Heatmap represents masked $H_{spec}$ scores of protease-substrate interaction. Arb. units, arbitrary units. Source data are provided as a Source data file.

under the curve (AUC) = 0.878 showing that the model exhibits strong discriminatory ability between true cleaving and non-cleaving pairs ($H_{spec}$ cutoff = -84) (Supplementary Fig. 3). These results demonstrate the predictive power of ProSSpeC and our ability to experimentally validate protease-substrate specificity in human cells.

Our experimental assay also revealed a subset of proteases that exhibited minimal or no cleavage-induced fluorescence (Supplementary Fig. 1). Two potential scenarios could explain the apparent lack of cleavage: (1) protease is autoinhibited, which has been shown previously for wild-type TEVp[20]; or (2) the putative substrate sequence does not promote cleavage[18]. We addressed the first scenario by mutating the residue position known to be involved in autoinhibition[20] (Supplementary Table 1). To address the second scenario, we retested nine proteases that exhibited minimal or intermediate cleavage efficiencies. Guided by the $H_{spec}$ score (Supplementary Table 2), substituting the triplet repeat of the 7-amino-acid substrate with the corresponding 20-amino-acid natural sequence resulted in a 1.3- to 7.9-fold increase in cleavage-induced fluorescence (Fig. 4a). While GS linkers flanking the 7-amino-acid motif enhanced cleavage in some cases, the 20-amino-acid natural context consistently produced higher activity. This result suggests that $H_{spec}$ scores improve when we include the biochemical context surrounding the substrate sequence, which translates to improved proteolytic activity.

**ProSSpeC-guided engineering of nonnative cleavage specificity**

To determine whether ProSSpeC can be used to predict protease mutations that improve or preserve catalytic efficiency, we mutated five proteases, including the widely used TEVp and TVMVp, with similar $H_{spec}$ scores to their wildtype counterparts. We found that these mutants have comparable efficiencies to wildtype (Supplementary Fig. 4), demonstrating the ability of ProSSpeC to explore sequence space using $H_{spec}$ as a proxy for functionality.

We then tested whether ProSSpeC can predict how single amino-acid changes in the substrate sequence affect protease activity. We calculated changes in $H_{spec}$ scores (DeltaH) for point mutations of target substrates that predict improved (DeltaH < 0) or reduced (DeltaH > 0) proteolytic cleavage. We assayed nine single-site mutations in target substrates and found that DeltaH was a good predictor of cleavage outcomes (as measured by changes in fluorescence intensity, DeltaFluor) (Supplementary Fig. 5). For example, mutating the ScMVcs P3 position PheTyr (DeltaH = −21.3) significantly improved the ScMVp-induced fluorescence in the assay (DeltaFluor = 0.54). By contrast, mutating the EAPVcs P2 LeuHis led to a large positive DeltaH (49.8) and significantly reduced EAPVp-induced fluorescence (DeltaFluor = −0.87). Furthermore, when comparing predicted increase or decrease of fluorescence across protease-substrate pairs, DeltaH is correlated with DeltaFluor ($R = −0.69$, $p = 9.2 \times 10^{-5}$) (Fig. 4b and Supplementary Table 3). These findings indicate that it is possible to model and predict

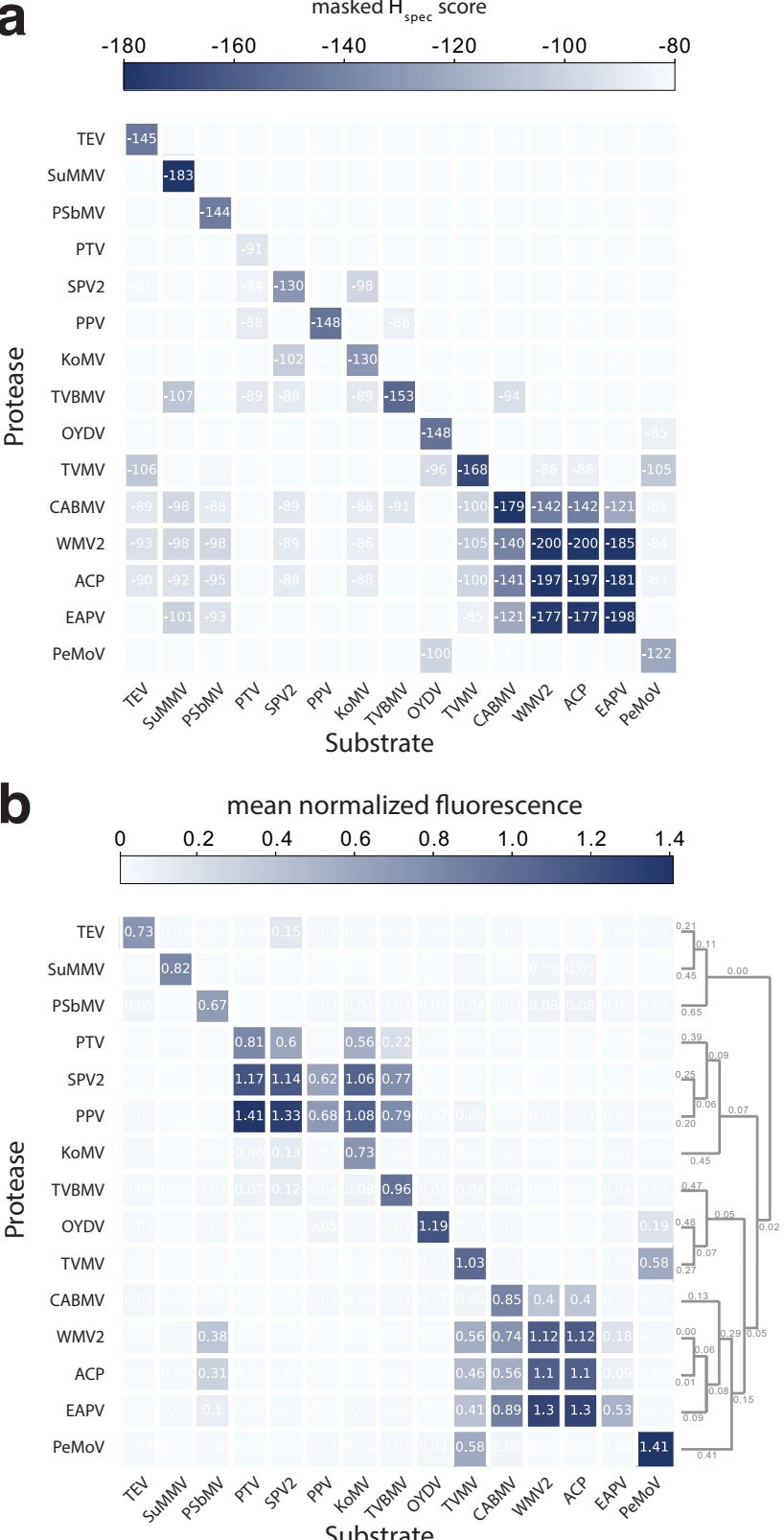

**Fig. 3 | Orthogonality of 225 protease-substrate pairs. a** As predicted by masked $H_{spec}$ scores and **b**, experimentally validated by normalized cleavage-induced fluorescence (data shown as mean values of $n = 3-5$ independent replicates). $H_{spec}$ and fluorescence values are correlated (Pearson $R = 0.68$; two-tailed $p = 1.40 \times 10^{-31}$) and have strong graph network similarity (DeltaCon similarity = 0.676). **b** *Right*: phylogenetic tree constructed using the maximum likelihood method in MEGA12, based on 15 protease amino acid sequences (245 aligned positions). Source data are provided as a Source data file.

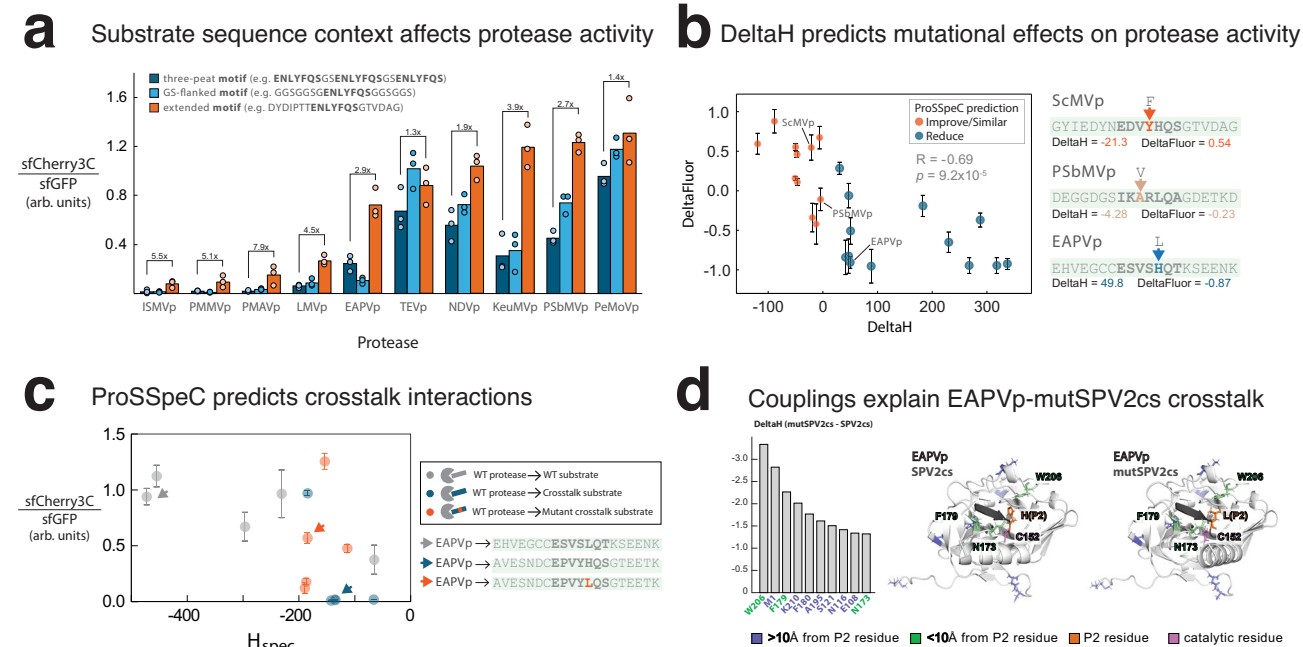

**Fig. 4 | Engineering protease specificity and protease-induced apoptosis.**
**a** Comparison of cleavage-induced fluorescence when an extended substrate context of 20 amino acids is used versus triple repeat of 7 amino acids and GS-flanked 7 amino acids. **b**, **c** ProSSpeC predicts the effects of substrate mutations on protease activity. **b** DeltaH ($\Delta H_{spec}$) versus DeltaFluor ($\Delta$ fluorescence) for each dot, which represents the difference between a protease cleaving one substrate vs. the same protease cleaving another substrate (see also Supplementary Table 3). $\Delta H_{spec}$ and $\Delta$ fluorescence are correlated (Pearson $R = -0.69$, one-tailed $p = 9.2 \times 10^{-5}$). **c** Cleavage-induced fluorescence of each protease tested against its cognate (WT) substrate, crosstalk substrate, or mutant crosstalk substrate, as a function of $H_{spec}$ score. For example, EAPVp was predicted to exhibit strong interaction with its

cognate substrate (gray), no crosstalk with SPV2cs (blue), and crosstalk with mutSPV2cs (orange). **d** Top 10 most negative coupling differences between EAPVp-SPV2cs and EAPVp-mutSPV2cs. *Left*, EAPVp-P2 residue pairs are arranged left to right in decreasing contribution to $H_{spec}$ score difference between the two protease-substrate pairs. *Middle* and *right*, Alphafold3 predicted structures of EAPVp with SPV2cs (*middle*) or with mutSPV2cs (*right*). Highlighted are EAPVp residues >10 Å away (violet), <10 Å away (green), P2 residue (orange), and catalytic residue (purple). In **a**, data are shown as mean values of $n = 3$ independent replicates and in **b**, **c** as mean values ± SEM of $n = 3$ independent replicates. Arb. units, arbitrary units. Source data are provided as a Source data file.

protease activity against different substrates at single-residue resolution.

### Engineering crosstalk for selective synpoptosis

Encouraged by the resolving power of ProSSpeC, we tested whether it could prescribe functional proteases for a substrate that undergoes a single amino acid mutation. For example, ProSSpeC prescribed that EAPVp can cleave the SPV2cs when mutated at the P2 site (EPVYHQ^-SEPVYLQ^S; DeltaH = −49.8) (Fig. 4c, Supplementary Fig. 6, and Supplementary Table 3). We experimentally observed this crosstalk (DeltaFluor = 0.60), which was surprising given that mutSPV2cs (EPVYLQ^S) is markedly different from the native EAPVcs (ESVSLQ^T) that EAPVp cleaves (Fig. 2a). To examine the basis of this prediction, we analyzed the top protease–substrate residue couplings contributing to the lower $H_{spec}$ score. The ten strongest couplings all involved the mutated substrate P2 residue and numerous EAPVp residue positions, which were distributed across the protease (Fig. 4d and Supplementary Fig. 7). These mutagenesis experiments clearly demonstrate that ProSSpeC can resolve the effect of single-site substrate mutations on protease activity and identify crosstalk interactions between proteases.

ProSSpeC allowed us to identify functional proteases and engineer their crosstalk at single-residue resolution, which opens the door to unique biotechnological applications. To demonstrate this potential, we tested whether proteases can be used to selectively kill cells[21,22] that harbor a point mutation in a target protein. We used and engineered Caspase3, an executioner caspase whose activation leads to cleavage of numerous endogenous proteins, ultimately leading to apoptosis. We tested whether EAPVp can synthetically trigger caspase-

mediated apoptosis (i.e., synopoptosis) of mutant cells in a mixed population with three cell types: a wild-type cell (expressing Caspase3 with a wild-type SPV2cs: EPVYHQS); a mutant cell (expressing Caspase3 with mutSPV2cs: EPVYLQS); or a cell with no Caspase3 (as negative control). Flow cytometry of the mixed population showed that EAPVp can selectively trigger apoptosis in cells that expressed mutant Caspase3, but not in cells expressing wild-type or no Caspase3 (Fig. 5a and see Supplementary Fig. 8 for gating strategy). As expected, EAPVp efficiently cleaved and activated EAPVcs-containing Caspase3. Similarly, SPV2p only triggered apoptosis in cells with Caspase3 harboring SPV2cs (Fig. 5b). This result demonstrates that engineered protease-substrate pairs can be used in biological circuits to detect and selectively trigger apoptosis of mutant cells in a mixed population.

### Discussion

We demonstrate that ProSSpeC, a quantitative model for predicting protease-substrate interaction, enables the identification and engineering of protease-substrate pairs with enhanced cleavage activity in human cells. Unlike most previous analyses of NIa protease activity[23–25], our ProSSpeC-guided experiments evaluated not only the 7-amino acid substrate but also its extended 20-amino acid context window, providing a more comprehensive assessment of protease-substrate specificity. Previous work by Beaumont et al.[26] showed that extended substrate context matters for cleavage specificity. Our family-wide analysis reveals that residues flanking the cleavage motif play a role in cleavage efficiency. Fig. 4a shows that the composition of amino acids (and not just length) outside of the P6-P1' region significantly improves catalytic efficiency (1.3 to 7.9-fold). This provides evidence that the extended context contributes to proteolytic cleavage. One potential

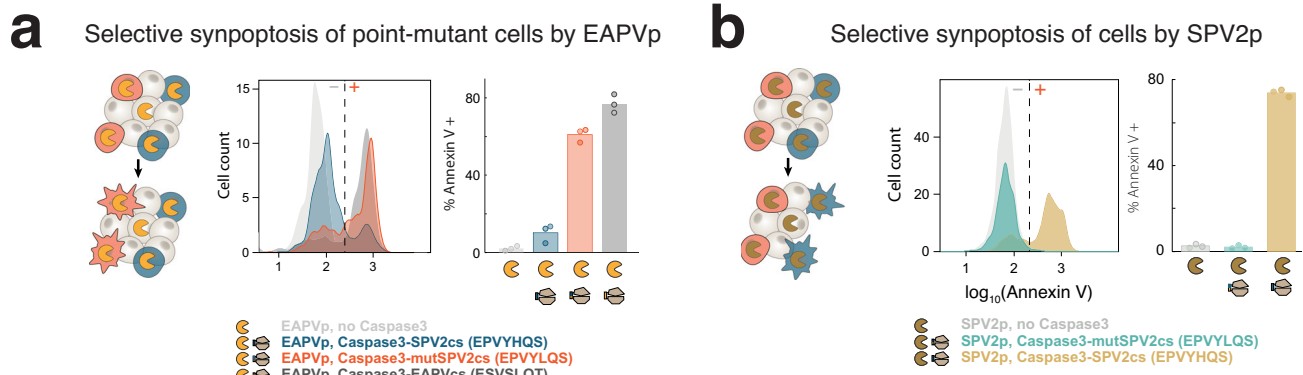

**Fig. 5 | Selective synoptosis of mutant cell types in a mixed population using a protease engineered to recognize the mutation.** Left: Schematic depicts selective synoptosis of **a** EAPVp and **b** SPV2p. Middle: flow cytometry histogram of Annexin V staining of the mixed cell population. Right: percentage of Annexin V+ cells by cell type. **a** Cell types expressed EAPVp (gray), EAPVp and SPV2cs-containing Caspase3 (blue), EAPVp and mutSPV2cs-containing Caspase 3 (orange), or EAPVp and EAPVcs-containing Caspase 3 (dark-gray). **b** Cell types expressed SPV2p (gray), SPV2p and mutSPV2cs-containing Caspase3 (teal), or SPV2p and SPV2cs-containing Caspase 3 (gold). Data are shown as mean values of $n = 3$ independent replicates. Source data are provided as a Source data file.

explanation may be that these regions have been subjected to evolutionary selection, potentially playing a role in the local structural dynamics of substrate recognition. Additionally, ProSSpeC can also predict cleavage activity at single-residue resolution (Fig. 4b, c), enabling programmable protease-substrate recognition. Compared to other generative design tools, ProSSpeC offers tunable cleavage specificity and amino acid-level interpretability with reliable success rates. We have developed a web application (https://coevolutionary.org/prosspec/) for researchers to explore the sequence space of NIa protease-substrate specificity.

To demonstrate the interpretability of the model, we analyzed a case where SPV2cs, which is not cleaved by EAPVp, was predicted to become cleavable upon a single amino acid substitution (Fig. 4c). We examined the top protease–substrate residue couplings contributing to this prediction. Of the ten strongest couplings, only three EAPVp residues were within 10 Å of the mutated substrate position (P2). The highest-ranked interaction was a hydrophobic contact between W206 and the mutant P2 Leu, whereas the wild-type P2 His, a positively charged residue, may be unfavorable for W206. The remaining seven top couplings involved residues located more than 10 Å from P2, suggesting that cleavage specificity arises partly from long-range epistatic interactions within a distributed residue network (Fig. 4d and Supplementary Fig. 7). These results illustrate how ProSSpeC captures interpretable residue–residue dependencies underlying substrate recognition.

One limitation shared by predictive models, including ProSSpeC, is the lower accuracy for sequences (or pairs of sequences) that significantly deviate from the natural distribution. For example, ProSSpeC typically predicts low specificity for substrates with non-glutamine amino acids at the highly conserved P1 site. Another example is the predicted promiscuity but experimentally high substrate specificity for CABMVp in Fig. 3. These results suggest that CABMVp may contain attributes that deviate from the learned NIa protease family properties. Normalizing noncognate $H_{spec}$ scores relative to the cognate $H_{spec}$ score enhanced discriminatory performance and decreased the incidence of false-positive predictions for CABMVp and other proteases. (Supplementary Fig. 3 and Supplementary Fig. 9). Still, to properly address this limitation, the low-homology sequence space needs to be experimentally explored. Retraining ProSSpeC on an expanded dataset could enhance its predictive power and expand the sequence diversity of engineered protease-substrate pairs. Nonetheless, we have shown that the current model is effective at engineering pairs for an expanded

distribution based on natural sequence properties, supporting immediate use and application of ProSSpeC.

While other models for protease cleavage exist, they focus on protease families other than NIa proteases[27–30], and more recently, only on TEVp[25]. By contrast, our ProSSpeC model provides full support for the entire NIa protease family (including TEVp) and leverages coevolutionary information to guide NIa protease-substrate specificity predictions, allowing for a comprehensive analysis of the underlying sequence space of both enzyme and substrate. We believe that our tool fills an important research niche that complements existing resources, providing computational support that did not previously exist.

We leveraged ProSSpeC to engineer proteases that alter cell phenotypes by selectively triggering apoptosis. Our results highlight the potential of engineered proteases to interface with endogenous cellular pathways; engineered proteases could be used to investigate protein function or as diagnostic tools to monitor changes in protein sequence or expression. The synergy between modeling and experimentation provides a platform to perform targeted protein editing at the cell proteome scale.

## Methods

### Homolog search and alignment

Protease sequences were obtained using HMMER[31]. The hidden Markov model profile (hmm seed) from the peptidase C4 family (InterPro PFAM accession: PF00863), which is the family that is used to describe NIa proteases, was used to run *hmmsearch* on the Uniprot databases (Sprot and TREMBL- February 2024 release). Sequences with greater than 45 contiguous gaps were removed from the alignment. The resulting protease alignment only contains sequences from *Potyviridae*. Substrate sequences were obtained by downloading the polyprotein sequences for the aligned and filtered proteases. To align the substrates such that the P1 position is at the same index, a manual alignment was created using the sequences denoted in Supplementary Table 4. The context for these sequences was extended to 20 amino acids based on previous findings[26], which showed that surrounding amino acids could induce TEVp cleavage in non-canonical consensus sequences. The resulting hmm profile was then used to align the polyprotein sequences to the manual alignment. Substrate sequence composition is shown in Supplementary Fig. 10. Then, the proteases and substrates originating from the same organism, as determined by Uniprot ID, were concatenated to each other to create the full alignment for specificity. This resulted in 3817 concatenated protease-substrate pairs, which are described in Supplementary Data 1. Another

alignment was created for nonspecific signals by randomly scrambling the proteases and substrates. Both alignments were then used to calculate parameters for the ProSSpeC model.

## Calculation and validation protease-substrate specificity calculator (ProSSpeC)

Direct coupling analysis (DCA) was performed using mfDCA[9] on the canonical pairs of proteases and substrates or the scrambled pairs. To check the quality of the mfDCA results, we plotted the top contacts based on direct information (DI) value as shown in Supplementary Figs. 11 and 12. The learned DCA parameters from the canonical pairs were then used to compute the Potts Hamiltonian value:

$$H_{Can} = -\sum_{i=1}^{L_{NIa}} \sum_{j=L_{NIa}+1}^{L_{NIa}+L_{Sub}} e_{ijCan}\left(A_i, A_j\right) - \sum_{i=1}^{L_{NIa}+L_{Sub}} h_{iCan}(A_i) \tag{1}$$

Where $H_{Can}$ is the Hamiltonian value for canonical pairings, $L_{NIa}$ is the length of the protease, $L_{Sub}$ is the length of the substrate, $e_{ij}$ is the coupling between amino acids at positions $i$ and $j$, and $h_i$ is the local field for the amino acid at position $i$. Then, the scrambled pairs were used to calculate the Potts Hamiltonian value for the nonspecific signal:

$$H_{Scram} = -\sum_{i=1}^{L_{NIa}} \sum_{j=L_{NIa}+1}^{L_{NIa}+L_{Sub}} e_{ijScram}\left(A_i, A_j\right) - \sum_{i=1}^{L_{NIa}+L_{Sub}} h_{iScram}(A_i) \tag{2}$$

The ProSSpeC model $H_{spec}$ score was then calculated by removing the nonspecific signal from the signal of the canonical pairs to yield the specific signal from the Hamiltonian:

$$H_{Spec} = H_{Can} - H_{Scram} \tag{3}$$

In Figs. 2b, 3, 4a; Supplementary Figs. 1 and 2 the substrate used in experimental assays was a triplet of the 7 amino acid consensus sequence. For these predictions, all substrate positions that are not P6-P1' are masked and delineated using masked $H_{spec}$:

$$e_{ij} = \begin{cases} 0 & \text{if } i \text{ or } j \in masked\ positions \\ e_{ij} & \text{otherwise} \end{cases} \quad \text{and} \quad h_i = \begin{cases} 0 & \text{if } i \in masked\ positions \\ h_i & \text{otherwise} \end{cases} \tag{4}$$

## $H_{spec}$ and fluorescence correlations and orthogonality metrics

Masked $H_{spec}$ values were clipped to the range [−180, −80], meaning that any score below −180 was set to −180 and any score above −80 was set to −80. This adjustment was made to make a fair comparison between fluorescence and $H_{spec}$ with the consideration that fluorescence intensity values cannot drop below 0. This cut-off is supported by the distribution of scrambled partners in comparison with natural partners. Masked $H_{spec}$ values were also multiplied by −1.0 to remove the inverse relationship, which is primarily important for the graph comparisons. In Fig. 3 correlation between masked $H_{spec}$ and fluorescence was calculated using Pearson $R$ across all cognate and crosstalk pairs ($R = 0.68$; $p = 1.40 \times 10^{-31}$). A graph network similarity for protease-substrate specificity was also calculated using DeltaCon[32]. A bipartite graph was constructed such that each protease and each substrate was represented as a node with edges between each protease node and each substrate node, yielding 30 nodes and 225 edges. Edge weights were set to the normalized masked $H_{spec}$ values for the $H_{spec}$ graph and the normalized fluorescence values for the experimental graph. Similarity was calculated to be 0.676 with a Matusita distance of 0.479 after running with the following parameters: random_seed = 42, $g = 20$, $\varepsilon = 0.01$.

## Phylogenetic tree construction

The WAG substitution model was used. Branch lengths, shown next to the branches (not drawn to scale), represent the number of substitutions per site. The initial tree for the heuristic search was selected by comparing neighbor-joining and maximum parsimony starting trees. Rate variation among sites was modeled using a discrete gamma distribution with four categories (+G, parameter = 1.8470), and 13.88% of sites were considered evolutionarily invariant (+I). Analyses were performed utilizing up to 12 parallel computing threads in MEGA12[33].

## Normalization of $H_{spec}$ orthogonality matrix

$H_{spec}$ scores were calculated for all pairwise protease−substrate combinations as previously described. To enable cross-protease comparison and reduce scale bias, raw $H_{spec}$ values were normalized relative to the corresponding diagonal (cognate-cleaving) interactions. For each element $H_{ij}$, the normalized orthogonality score was computed by normalizing with respect to the cognate (diagonal) term $H_{ii}$:

$$H_{ij}^{norm} = \frac{H_{ij}}{H_{ii}} \tag{5}$$

This yielded a complementary matrix−normalized (by division) (Supplementary Fig. 9)−representing relative specificity between non-cognate and cognate cleavage pairs. The matrix was visualized as heatmaps and subsequently used as input predictors for ROC analysis.

## ROC analysis and performance metrics

Protease−substrate matrices (15 × 15) of predictors ($H_{spec}$ variants) and experimental readouts (fluorescence) were vectorized by row-major flattening to obtain paired lists of scores and labels. Ground-truth labels were defined from fluorescence using a prespecified threshold: positive (cleaving) if fluorescence ≥ 0.1. Because lower raw $H_{spec}$ values indicate stronger matches, raw $H_{spec}$ scores were sign-inverted prior to analysis (i.e., we used−$H_{spec}$); normalized $H_{spec}$ were used as-is. Receiver operating characteristic (ROC) curves were computed in Python (v3.11) using *scikit-learn* (roc_curve, auc; v1.4). For each distinct score threshold, the true-positive rate (TPR) and false-positive rate (FPR) were calculated as:

$$TPR = \frac{TP}{TP+FN}, \ FPR = \frac{FP}{FP+TN} \tag{6,7}$$

where TP, true positive; FP, false positive; TN, true negative; FN, false negative. The area under the ROC curve (AUC) was obtained via trapezoidal integration (auc). An optimal decision threshold on the predic-tor was chosen by maximizing Youden's J (TPR−FPR) across all thresholds.

## Cleavage-induced fluorescence assay

We fused a cytoplasmic protein, arrestin beta-2, and sfCherry3C(1-10) (i.e., the first ten beta strands of the sfCherry3C fluorescent protein), such that cleavage of the latter allows it to bind with the eleventh beta strand sfCherry11M (fused to the histone H2B-sfGFP reporter) and reconstitute sCherry3C fluorescence[34]. Measurements for protease-induced reconstitution of sCherry3C fluorescence were normalized to the reporter sfGFP fluorescence. To prevent previously known autoinhibition of potyviral proteases[17], we mutated serine in position 219 (in TEVp) to valine when present (see Supplementary Table 1 for homologous mutations in other proteases).

To benchmark the accuracy of the fluorescent protein reconstitution assay, the 31 pairs were also tested, where each protease (and the nuclear reporter) was tagged to sfGFP and each substrate to mTagBFP2. Cells were then analyzed through flow cytometry (BD LSRFortessa™ Cell Analyzer). Cleavage-induced fluorescence intensities normalized against protease amount, substrate amount, or both

yielded strong correlations (Pearson $r = 0.8$ ($p = 1.8 \times 10^{-26}$), 0.82 ($p = 1.5 \times 10^{-28}$), and 0.81 ($p = 7.4 \times 10^{-27}$), respectively) with the previous assay, suggesting that both assays reflect the actual efficiency of proteolysis (Supplementary Fig. 13).

## Cloning

All plasmids use a pCS2+ backbone and were constructed using either Gibson assembly or site-directed mutagenesis (SDM) techniques (Supplementary Data 2). Protease sequences were synthesized (Integrated DNA Technologies, IDT), amplified using Phusion® High-Fidelity PCR Kit (New England Biolabs, NEB), and inserted into the pCS2+ backbone using In-Fusion® Snap Assembly Master Mix (Takara Bio). For point mutants, SDMs were performed using the Q5® Site-Directed Mutagenesis Kit (NEB). Oligonucleotide primers were also synthesized by IDT.

## Cell culture

Human Embryonic Kidney (HEK) 293 cells (RRID:CVCL_0045) were a gift from the lab of Dr. Leonidas Bleris. Cells were cultured in a humidity-controlled incubator under standard culture conditions (5% $CO_2$, 37 °C) using complete growth media consisting of Dulbecco's High Glucose Modified Eagles Medium (DMEM high glucose–Cytiva; catalog#SH30022FS), supplemented with 10% fetal bovine serum (FBS–Corning; catalog#MT35011CV), 1% penicillin–streptomycin (Corning; catalog#MT30002CI), and 1X MEM non-essential amino acid solution (NEAA–Sigma-Aldrich; catalog#M7145). Cells were passaged at 70–90% confluency. For subculturing, cells in T75 flasks were added with 3 mL of trypsin-ethylenediaminetetraacetic acid (EDTA) (Gibco; catalog#25200056) and incubated for 5 min at 37 °C. To inactivate trypsin, 5 mL growth media was added. Cells were spun down (130 × $g$, 5 mins), supernatant discarded, and cell pellet resuspended in 10 mL of fresh growth media. A 1 mL aliquot of the cell suspension was added for each T75 flask (Corning; catalog#07202000) containing 10 mL of growth media for a 1:10 split ratio.

## Transfection, imaging, and analysis

One day before transfection, HEK293 cells were seeded in 24-well plates at 75,000 cells/well. Transfections were done with the jet-PRIME® reagent (Sartorius), following their recommended protocol. Each well was transfected with a ratio of 1:10:1 of protease:substrate:reporter (16 ng each of reporter and protease, 166 ng of substrate, and 50 ng of mock plasmid DNA not encoding any protein–adding up to 250 ng total DNA) mixed with 0.5 mL of jetPRIME® reagent in 25 mL of jetPRIME® buffer. All transfections were done this way except for Fig. 3b; where the same ratio of protease:substrate:reporter was followed except 300 ng of mock was added instead of 50 ng (adding up to 500 ng of total DNA), which were then mixed with 1 mL of jetPRIME® reagent in 50 mL of jetPRIME® buffer. Images were taken 24 h after transfection using EVOS M5000 fluorescence microscope with a 20× objective (NA:0.45, WD:6.12 mm, Cat. no: AMEP4982). Fluorescence images were analyzed using FIJI. Fluorescence intensity values were calculated by subtracting background fluorescence from nuclei fluorescence and then normalizing output fluorescence (sfCherry3C) by dividing it with reporter fluorescence (sfGFP). For each condition, three biological replicates with two technical replicates each were done (corresponding to two wells). For each technical replicate, 30 nuclei were analyzed.

## Flow cytometry

Media from each well were discarded, and cells in each well were then detached by adding 250 µL of trypsin-EDTA and placed in the incubator (37 °C, 5% $CO_2$) for 5 min. Wells containing trypsin-detached cells were added with equal amount of fresh media to stop the reaction. Cells were transferred to 1.5 mL tubes, spun down (130 × $g$ for 5 min), washed once with ice-cold PBS, spun down again (130 × $g$ for 5min),

and resuspended in 500 µL DMEM high glucose + 2% FBS. Cells were transferred to 5-mL round-bottom tubes through the cell strainer caps (352235, Falcon®) before running flow cytometry using the BD LSRFortessa™ Cell Analyzer. A minimum of 10,000 events were analyzed per condition.

## Protease titration

Using the same transfection and analysis protocol, we tested five proteases across a range of plasmid concentrations (1.66, 5, 16.6, and 50 ng) to evaluate their proteolytic efficiency. As protease expression increased, some proteases showed higher activity while others reached an early plateau (Supplementary Fig. 14). Notably, all proteases reached fluorescence saturation at 16.6 ng, the concentration used throughout this study. The resulting fluorescence, therefore, reflects apparent catalytic efficiencies that can be directly compared across proteases.

## Structural modeling

We modeled the protease in complex with the corresponding 7-amino-acid substrate using AlphaFold3 (alphafoldserver.com)[15] to generate high-confidence structural predictions. The amino acid sequences of the protease and substrate were input as two different protein entities to capture protease-substrate interactions. The resulting top-ranked structural models were visualized using PyMOL (Version 3.1.3) to highlight the top ten couplings as well as the catalytic residue for each protease-substrate pair in Fig. 4d.

## Selective synpoptosis of mutant cells

Caspase 3, whose natural cleavage site was replaced with either SPV2cs or mutSPV2cs, was co-transfected with either SPV2p or EAPVp into HEK293 cells. After 12 h, cells were then stained with an apoptosis marker fluorophore-conjugated Annexin V, and the immunofluorescence intensities were measured using flow cytometry.

## Synpoptosis transfection

One day before transfection, HEK293 cells were seeded in a 6-well plate at 300,000 cells/well. To generate a mixed population of cells transfected with a single Caspase3 type (EAPVp-cleavable Caspase3 tagged with mCherry; or SPV2p-cleavable Caspase3 tagged with sfGFP), liposome complexes were individually made for each plasmid. In Caspase3-only conditions, 20 ng of EAPVp-cleavable Caspase3 or SPV2p-cleavable Caspase3 was mixed with 230 ng of mock plasmid. Each mixture was then incubated with 1 µL of jetPRIME reagent in 50 µL of jetPRIME buffer. In conditions containing SPV2p and protease-cleavable Caspase3, 20 ng of EAPVp-cleavable Caspase3, 20 ng of SPV2p-cleavable Caspase3, or 60 ng of SPV2p-mTagBFP2 were separately mixed with mock plasmid to a total of 166 ng for each two-component mixture. The same procedure was done with conditions containing EAPVp-mTagBFP2 (120 ng). Each mixture was then incubated with 0.66 µL of jetPRIME reagent in 33.3 µL of jetPRIME buffer. After a 10-min incubation, the complexes for each condition were added to each well. This procedure ensured that each cell in a mixed population expressed only either EAPVp-cleavable or SPV2p-cleavable Caspase3.

## Synpoptosis staining and flow cytometry

Media from each well were individually collected into a 1.5-mL centrifuge tube. Cells in each well were detached using 1 mL of trypsin-EDTA for 5 min. Trypsin-detached cells were transferred to previously collected media to stop the reaction. Cells were spun down (130 × $g$ for 5 min), resuspended in fresh media, and placed in the incubator (37 °C, 5% $CO_2$) for 30 min to allow recovery. They were then spun down again, washed once with ice-cold PBS, and resuspended in Annexin V buffer (600 µL/well) with Annexin V stain (30 µL/well). Alexa Fluor 350-conjugated Annexin V (Cat. no. A23202, Invitrogen) was used for

Caspase3 titration (see Supplementary Table 5) while Alexa Fluor 647-conjugated Annexin V (cat. no. A23204, Invitrogen) was used for the synopoptosis experiment (see Fig. 5 and Supplementary Fig. 8). Annexin V staining was done at 25 °C for 15 min. Cells were then transferred to round-bottom tubes through a cell strainer cap before running flow cytometry using the BD LSRFortessa™ Cell Analyzer. Gating shown in Supplementary Fig. 8. A total of 100,000 events were analyzed per condition.

## Software

Protease and substrate sequences were gathered using Matlab 2023b and hmmer 3.4. In silico plots were generated usingPython 3.11.5 (scipy 1.11.1, matplotlib 3.7.2, biopython 1.78, pandas 2.1.1, numpy 1.24.3, ete3 3.1.3). Fluorescence microscopy images were analyzed with ImageJ software (v 2.14.0/1.54 f). For all flow cytometry experiments, flow cytometry data were processed using FlowJo software (v10.10.0). All figure schematics were generated or compiled using Adobe Illustrator (v. 29.6.1).

## Statistics and reproducibility

No statistical method was used to predetermine sample size. No data were excluded from the analyses. The experiments were not randomized. The investigators were not blinded to allocation during experiments and outcome assessment.

All experiments were performed in biological triplicates, except for Fig. 3b as noted performed with $n = 3-5$ replicates. Where error bars are shown, results were also expressed as means ± SEM. Correlations were evaluated using Pearson's correlation coefficient ($r$). Two-tailed (Fig. 3) or one-tailed (Fig. 4b and Supplementary Fig. 13) $p$-values were calculated, and correlations were considered statistically significant at $p < 0.05$.

## Reporting summary

Further information on research design is available in the Nature Portfolio Reporting Summary linked to this article.

## Data availability

The data generated in this study have been deposited in the Zenodo database under accession code https://doi.org/10.5281/zenodo.15039890. The overview of aligned *Potyviridae* sequences and the list of plasmids (Supplementary Data 2) used in this study are provided as Supplementary Data. The plasmid sequences and maps for all proteases, the 7-, GS-flanked 7-, and 20-amino acid substrate of TEVp, and the H2B-sfGFP reporter used here are available on Addgene: https://www.addgene.org/Dave_Dingal/. Source data are provided with this paper. The protein structural data used in this study are available in the PDB database under accession code 1LVM. The protein family profile HMM used in this study are available in the InterPro database under accession code PF00863. Source data are provided with this paper.

## Code availability

To facilitate the testing of thousands of potyviral proteases against peptide targets by multiple laboratories, we created an interactive web application for ProSSpeC (https://coevolutionary.org/prosspec/). Code is available at https://github.com/morcoslab/ProSSpeC and archived on Zenodo with https://doi.org/10.5281/zenodo.18321025[35].

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

## Acknowledgements

We thank members of the Dingal lab and Morcos lab for their advice, expertise, and discussions. We thank Elliott Joe, Ahmed Adookkattil, and Shashwat Singh for data analysis support. We also thank the UTD Flow Cytometry Core for infrastructure and support. We acknowledge the UTD Office of Information Technology Cyberinfrastructure Research Computing for providing high-performance computing and services. M.B.L. is supported by the UTD Eugene McDermott Graduate Fellowship. This research was supported by a UTD Startup Fund and National Institutes of Health-NIGMS awards to the labs of P.C.D.P.D. (R35GM150967) and of F.M. (R35GM133631). F.M. acknowledges support from the National Science Foundation (MCB-1943442).

## Author contributions

M.B.L., Z.S., A.S.Y., A.T., R.R., J.N., and J.K. performed experiments. M.B.L. and P.C.D.P.D. analyzed experimental results. Computational modeling and full stack development of the ProSSpeC web app performed by C.Z. Conceptual planning and resources provided by P.C.D.P.D. and F.M. M.B.L., C.Z., F.M., and P.C.D.P.D. authored and edited this manuscript, including illustrations. The final version of this manuscript is approved by all authors.

## Competing interests

The Board of Regents of The University of Texas System have filed a pending patent application on behalf of co-inventors P.C.D.P.D., F.M., C.Z., and M.B.L. of the engineered proteases described (US Provisional Application No. 63/885,099). The remaining authors declare no competing interests.

## Additional information

**Supplementary information** The online version contains Supplementary material available at https://doi.org/10.1038/s41467-026-69961-5.

