## [Peer Review File · Nature Communications]

Identification and engineering of highly functional Potyviral proteases in cells using co-evolutionary models

Corresponding Author: Professor P. C. Dave Dingal

Version 0:

Reviewer comments:

Reviewer #1

(Remarks to the Author)

In this work, the authors present ProSSpeC (Protease Substrate Specificity Calculator), a coevolution-based statistical model derived from Direct Coupling Analysis (DCA) to predict the substrate specificity of Potyviridae N1a proteases. The authors built this model from over 3800 naturally paired protease-substrate sequences and used the Hamiltonian-based Hspec score to quantify specificity.

They validated the model through a fluorescent protein reconstitution assay in human cells, demonstrating a significant correlation between the model-predicted Hspec scores and observed proteolytic activity. Beyond validating native protease-substrate pairs, they applied the model to: Predict the impact of single-residue mutations on substrate cleavage; engineer orthogonal and crosstalk interactions; construct a synthetic protease-triggered cell death pathway (synoptosis) that discriminates between mutant and wild-type cells.

This manuscript is rigorously constructed, starting from a clear guiding hypothesis, validated through extensive experimental assays, and expanded into multiple applications. It provides a conceptually solid framework that integrates bioinformatics and statistical mechanics, distinguishing itself from many black-box AI approaches in the field. The ProSSpeC model is elegant, interpretable, and practical, with promising implications for synthetic biology, therapeutic design, and enzyme engineering. I am supportive of the publication of this paper after addressing the following points:

a. Clarify the Hspec model in the Results section

Currently, the formulation and rationale of the Hspec model are only discussed in the Methods. A more detailed and accessible explanation of the model, including how the joint Hamiltonian is derived and why it works, should be included in the Results. This would allow readers to grasp the mechanistic intuition and significance of the model early on.

b. Clarify the correlation analysis

The manuscript mentions a strong correlation between Hspec scores and cleavage-induced fluorescence ($R = 0.79$), but it's unclear how this correlation was calculated. Was every protease-substrate pair (including poorly active or non-cognate combinations) included in the analysis, or was the correlation assessed only on best-performing pairs? Please clarify the subset used and justify the approach.

c. Explore mechanistic insight via structure-based modeling

While the model performs impressively, the molecular mechanism underlying specificity remains abstract. I suggest that the authors select one representative protease-substrate pair (perhaps one with a well-characterized Δ Hspec mutation) and perform structural modeling (e.g., docking or AlphaFold3-based complex prediction). This could shed light on how specific residue-residue interactions enhance or weaken specificity, thereby linking statistical couplings to physical mechanisms.

d. Discuss out-of-cleavage-site effects

In the Discussion, the authors should comment on the limitation of focusing solely on the cleavage region. Mutations outside the P6–P1' region (e.g., distal regions in the substrate protein) may alter local structure or dynamics, affecting cleavage. Acknowledging this limitation and suggesting future directions for extending the model (e.g., through structural context or experimental feedback loops) would strengthen the discussion.

e. Address substrate promiscuity

ProSSpeC is designed around specificity prediction, but some proteases may exhibit promiscuous behavior—cleaving multiple substrates with moderate affinity. Could the authors comment on how the model handles such cases? Does a moderately negative Hspec score reliably indicate partial activity across multiple substrates?

f. Clarify the CABMV case

The CABMV protease shows high cleavage efficiency toward its own substrate in the assay, but the Hspec score suggests potential promiscuity. This case deserves further discussion—does it suggest a decoupling between experimental signal

strength and model specificity? Could this be a case of an energetically favorable but non-unique interaction? Understanding and explaining such discrepancies will help delineate the boundaries of the model.

(Remarks on code availability)

I failed to access the code as it shows "this connection is not private".

Reviewer #2

(Remarks to the Author)

(Remarks on code availability)

Clear and easy to use

Reviewer #3

(Remarks to the Author)

In the manuscript "Identification and engineering of highly functional synthetic proteases in cells using co-evolutionary models," the authors present Protease Substrate Specificity Calculator (ProSSpeC), a co-evolution-guided computational tool that predicts protease-substrate specificity. Using direct coupling analysis, ProSSpeC quantifies protease-substrate specificity via a Hamiltonian score. ProSSpeC was trained on over 3,000 natural Nla protease-substrate pairs from the Potyviridae family. It accurately predicted cleavage efficiency for many protease-substrate pairs, with Hamiltonian scores strongly correlated with experimental cleavage-induced fluorescence in human cells. Several proteases were found to outperform the commonly used TEV protease (TEVp) in human cells. Additionally, ProSSpeC predicted the effects of single-point mutations in the cleavage sites of substrates, enabling tuning of substrate specificity. The authors further used ProSSpeC to reprogram protease specificity and engineer crosstalk between a protease and a mutant non-cognate substrate. As a demonstration, they used a protease to selectively trigger apoptosis in cells expressing engineered caspase-3 containing the mutant cleavage site of another protease.

Overall, this manuscript presents a co-evolutionary computational model supported by cell-based experimental validation. The model quantitatively predicts protease-substrate specificity at single-residue resolution. Integration of modeling with cell-based assays establishes a 'predict-validate' story. This study can offer programmable protease tools in synthetic biology, potentially enabling therapeutic applications such as selective cell killing.

Major Comments:

In the cellular fluorescence assay, it would be helpful if both the protease and the substrate were tagged with different fluorescent markers. That way, expression levels of both components can be monitored and controlled for by flow cytometry gating. Without rigorous quantification, it is difficult to tell whether differences in activity are really due to actual proteolysis efficiency or just variable expression or stability. Some data on protease and substrate levels would be necessary.

In the "ProSSpeC-guided engineering of nonnative cleavage specificity" section, it would add value to the paper if the authors systematically profile a library of substrate mutations for TEVp and TVMVp, which are widely used. Knowing how to improve cleavage efficiency for these proteases through targeted mutations would be valuable for many synthetic biology researchers. Right now, the focus is on ScMVp, SbMVp, and EAPVp, which are less common and could limit potential protease-based applications.

ProSSpeC offers quantitative Hamiltonian scores, but does not provide mechanistic insight or structural rationale for the most favorable protease-substrate pairs. Consider using AlphaFold-predicted complex structures or contact maps to rationalize top-scoring interactions.

The manuscript does not compare ProSSpeC against alternative protease-substrate specificity predictors, such as PROSPER, PeptideCutter, and iProt-Sub. A side-by-side comparison, even just for a subset of known pairs, would strengthen the claim that coevolutionary modeling adds value.

The observation that extending substrate sequences from 7 to 20 amino acids improves cleavage is interesting, but it is only shown for a few cases. Also, it would be good to clarify what the additional 13 amino acids are. Are they from natural sequence context, or are they synthetic linkers? Could flexible sequences like GGS repeats also boost cleavage? Would the position of the 13 amino acids matter, such as on the N-terminus or the C-terminus of the 7-amino acid cleavage site?

The fluorescence signal for some protease-substrate pairs seems to plateau, potentially masking finer distinctions. It would be helpful to report absolute fluorescence ranges and dynamic range limits for the assay. Consider using titration experiments to better quantify proteolytic efficiency.

The ProSSpeC model is validated using a relatively limited number of 10×10 protease-substrate pairs. It would be great to see more pairs tested to better assess how generalizable the model is.

Minor Comments:

Can the authors comment on how this tool can or cannot be applied beyond the Potyviridae family? Would the same modeling approach generalize to other protease families?

It would be helpful to indicate where TEV protease lands on the distribution in Figure 2a—TEVp is a standard point of comparison, so readers will naturally look for it.

Many experiments used only two independent replicates. N = 3 or more would be ideal.

In Figure 4b, it is not entirely clear what each dot represents. Clarifying that would help readers interpret the plot.

Figure 4d could benefit from a few more control conditions where the cell expresses “EAPVp + Caspase3-EAPVcs” and “SPV2p + Caspase3-SPV2cs”.

Define synoptosis more clearly on first mention.

Lastly, the authors wrote: “Our results highlight the potential of engineered proteases to interface with endogenous cellular pathways.” However, there is no demonstration of such interfacing in this study.

(Remarks on code availability)

n/a

Reviewer #4

(Remarks to the Author)

Here the authors developed ProsspeC, a generative model that was trained using 3817 pairs of viral Nuclear Inclusion a (NIa) proteases and their cleavage peptide sequences. The model can generate a score (Hspec) for an input, with lower scores (negative scores) predicts higher chance of cleavage. The authors then validated this model using a cell-based assay based on reconstitution of fluorescence protein (successful cleavage releases a fragment, which enters the nucleus and complements the other fragment in the nucleus to form a complete fluorescent protein).

The writing is exceedingly short. This is not acceptable. The whole paper reads like a summary, instead of a full-length scientific paper. Lack of details renders it difficult to review this paper as many figures and authors' interpretations are difficult to understand.

The study is limited to NIa protease family. No other types of proteases are examined. Authors need to tune down all broad claims of “proteases” and spell out clearly that the work is limited to NIa proteases in the title and abstract.

There is no “engineer protease” in this manuscript. Authors only tested mutated substrate sequences. The last paragraph of the “Discussion” needs to be corrected.

It is very strange that Fig. 1-4 are all covered by the first sub-title of the “Result” section, and the main figures only have these 4 figures. Many supplementary figures should be moved to main figures.

There are many cases that experimental data do not fit their Hspec prediction. Authors need to explain all these pairs that do not fit their models: Fig. 2b: SPV2p? PTVp? Fig. 3b: PPV versus KoMV substrate? Fig. S2: TEV: the score of -249 showed higher signal than -296. For PeMoV, -243 showed lower signal than -231? Fig. S3: for ScMV: the blue (-184) was predicted to be stronger than orange (-153), but the sfCherry3C signaling is higher for orange, how do you explain this? Fig. 4a: increase length to 20 did not drastically increase the degree of cleavage for ISMVp etc.

Fig. 3a: aren't these pairs used to train the model? Why it is a surprise that they correlate?

The meaning of Fig. 4d is not clear: there is no engineering of protease. The authors just put a mutant substrate peptide sequence into caspase3. In this case, wouldn't it be better to use the native substrate of this protease (EAPVp)?

(Remarks on code availability)

Version 1:

Reviewer comments:

Reviewer #1

(Remarks to the Author)

The authors have addressed all my critiques in the revised manuscript. I am supportive of publication.

(Remarks on code availability)

Reviewer #3

(Remarks to the Author)

The revision addresses all of our concerns.

(Remarks on code availability)

Summary Response to Reviewers:

We thank the Reviewers for their constructive feedback on the manuscript. We performed substantial, additional experiments, modeling, and clarifications on some conceptual and experimental details in the text. We summarize the new results that further clarify or strengthen our previous claims:

- (1) In updated Fig. 3, we expanded the protease-substrate orthogonality matrix from 100 to 225 experimentally tested pairs. ProSSpeC H_{spec} scores remained strongly correlated to the expanded experimental dataset of on-target and crosstalk activity. Further, to rigorously quantify the predictive capacity of H_{spec} scores, we generated receiver operating characteristic (ROC) curves in new Supplementary Fig. S2. Raw or normalized H_{spec} scores (AUC = 0.878; normalized AUC = 0.905) below a raw H_{spec} cutoff of -84 can reliably discriminate between true cleaving vs non-cleaving pairs. We added a new section in Methods to describe the ROC analysis.
- (2) In updated Fig. 4a, additional experiments on the substrate sequence context support the ProSSpeC prediction that the composition of the extended 20-a.a. substrate sequence (and not just linker length) substantially improves proteolytic activity for this family of proteases when compared to the minimal 7-a.a. substrate motif.
- (3) In new Supplementary Fig. S3, we experimentally validated 12 additional protease triple-mutants predicted by the ProSSpeC model to have similar or more negative H_{spec} scores than their wildtype counterparts. These mutants exhibited cleavage efficiencies similar to wildtype. We note that mutations can easily destroy protein function, but here we show that ProSSpeC-guided mutations efficiently maintain the enzyme-substrate specificity relationship, allowing one to explore sequence space using the H_{spec} score as a predictor of functionality.
- (4) In new Fig. 4d, to offer possible structural insights on ProSSpeC predictions, we also performed AlphaFold3 modeling to visualize how EAPVp effectively cleaves a non-canonical target (mutSPV2cs: SPV2cs with its P2 site mutated H → L). The top 50 amino-acid P2-couplings that contribute to a more favorable H_{spec} score for EAPVp-mutSPV2cs (compared to EAPVp-SPV2cs) are found in new Supplementary Fig. S6. We mapped the top 10 contributing residues of EAPVp onto AlphaFold3 models of (non-cleaving pair) EAPVp-SPV2cs or (cleaving pair) EAPVp-mutSPV2cs complexes. A distributed network of both active-site and epistatic residues contributes to the observed EAPVp-mutSPV2cs crosstalk, which highlights the single-amino acid resolution and *direct interpretability* of ProSSpeC model predictions.
- (5) In new Supplementary Fig. S11, we benchmarked and validated our cleavage-induced fluorescence assay against new flow cytometry data from fluorescent protein-tagged protease and substrate, for all 31 cognate pairs tested. Flow cytometry benchmarking further strengthened the robustness of the previous fluorescence image analyses (Pearson $r > 0.80$). We thoroughly described this benchmarking assay in Methods.
- (6) In new Supplementary Fig. S12, we also performed protease titration experiments to demonstrate that all proteases reached fluorescence saturation at the protease amount used throughout this study, such that readers can directly compare cleavage-induced fluorescence values between proteases.
- (7) All experiments were performed with independent replicates $N = 3-5$.

We have clarified the text where needed. We wrote specific responses to the Reviewers, as follows:

Specific Responses to Reviewers:

Reviewer #1 (Remarks to the Author):

In this work, the authors present ProSSpeC (Protease Substrate Specificity Calculator), a coevolution-based statistical model derived from Direct Coupling Analysis (DCA) to predict the substrate specificity of Potyviridae NIa proteases. The authors built this model from over 3800 naturally paired protease-substrate sequences and used the Hamiltonian-based Hspec score to quantify specificity. They validated the model through a fluorescent protein reconstitution assay in human cells, demonstrating a significant correlation between the model-predicted Hspec scores and observed proteolytic activity. Beyond validating native protease-substrate pairs, they applied the model to: Predict the impact of single-residue mutations on substrate cleavage; engineer orthogonal and crosstalk interactions; construct a synthetic protease-triggered cell death pathway (synoptosis) that discriminates between mutant and wild-type cells. This manuscript is rigorously constructed, starting from a clear guiding hypothesis, validated through extensive experimental assays, and expanded into multiple applications. It provides a conceptually solid framework that integrates bioinformatics and statistical mechanics, distinguishing itself from many black-box AI approaches in the field. The ProSSpeC model is elegant, interpretable, and practical, with promising implications for synthetic biology, therapeutic design, and enzyme engineering.

Response 1.1:

We thank the Reviewer for supporting our work and appreciate the thoughtful comments and suggestions. We hope your specific concerns are sufficiently addressed in this new revision, as discussed point-by-point below.

Reviewer #1:

I am supportive of the publication of this paper after addressing the following points:

a. Clarify the Hspec model in the Results section

Currently, the formulation and rationale of the Hspec model are only discussed in the Methods. A more detailed and accessible explanation of the model, including how the joint Hamiltonian is derived and why it works, should be included in the Results. This would allow readers to grasp the mechanistic intuition and significance of the model early on.

Response 1.2:

Details regarding the H_{spec} model were added to the Results section, along with the rationale of its significance. We hope that the following changes provide readers with a stronger intuition of the model as they read through the Results:

“ProSSpeC is a coevolutionary model that leverages the covariation of NIa protease sequences along with their aligned substrate sequences to calculate a Hamiltonian specificity score (H_{spec}), where the more negative the H_{spec} score is, the stronger the predicted specificity is. This specificity score is defined by the Potts Hamiltonian energy of canonical protease-substrate pairs (Equation 1), with the Potts Hamiltonian energy of scrambled enzyme-substrate pairs (Equation 2) removed, to provide a score of sequence attributes unique to interacting pairs and not shared due to family-likeness or general sequence similarity (Equation 3). For both canonical and scrambled protease-substrate pairs, the Potts Hamiltonian is defined as the sum of all parameters in a Boltzmann distribution, with couplings (e_{ij}) and local fields (h_i) inferred using direct coupling analysis (DCA), specifically the mean-field formulation (mfDCA)⁹. A similar rationale for using the DCA Hamiltonian to characterize molecular interactions has been applied to study specificity in two-component systems¹², recognition in protein-RNA

interactions¹⁷, and to predict and engineer compatibility in hybrid transcription factors^{13,18}. The ProSSpeC model applies this approach to quantify interactions of the protease and the 7-amino acid substrate; we masked out substrate residues outside of this window in H_{spec} calculations to yield the masked H_{spec} (Equation 4). The masked H_{spec} was utilized for predictions where the designed experiment did not include the extended substrate context. ProSSpeC allows us to quantitatively assess the specificity of Nla proteases and their substrate motif through sequence alone, allowing us to filter and rank both natural Nla protease-substrate pairs as well as mutated ones prior to experimental testing."

Reviewer #1:

b. Clarify the correlation analysis

The manuscript mentions a strong correlation between Hspec scores and cleavage-induced fluorescence (R = 0.79), but it's unclear how this correlation was calculated. Was every protease-substrate pair (including poorly active or non-cognate combinations) included in the analysis, or was the correlation assessed only on best-performing pairs? Please clarify the subset used and justify the approach.

Response 1.3:

The correlation calculation in this study took into account all the protease-substrate combinations that we had tested; there was no pre-selection. In the Methods section, we have improved the language to assist the reader in understanding the correlation calculation and added the following text to clarify our approach. Additionally, in response to Reviewer 3, we now have more than doubled the number of protease-substrate pairs that were experimentally tested (in new Fig. 3; 225 total pairs; Pearson R = 0.68).

H_{spec} and fluorescence correlations and orthogonality metrics

Masked H_{spec} values were clipped to the range [-180,-80], meaning that any score below -180 was set to -180 and any score above -80 was set to -80. This adjustment was made to make a fair comparison between fluorescence and H_{spec} with the consideration that fluorescence intensity values cannot drop below 0. This cut-off is supported by the distribution of scrambled partners in comparison with natural partners. Masked H_{spec} values were also multiplied by -1.0 to remove the inverse relationship, which is primarily important for the graph comparisons. In **Fig. 3** correlation between masked H_{spec} and fluorescence was calculated using PearsonR across all cognate and crosstalk pairs (R = 0.68; $p = 1.40 \times 10^{-31}$). A graph network similarity for protease-substrate specificity was also calculated using DeltaCon³². A bipartite graph was constructed such that each protease and each substrate was represented as a node with edges between each protease node and each substrate node, yielding 30 nodes and 225 edges. Edge weights were set to the normalized masked H_{spec} values for the H_{spec} graph and the normalized fluorescence values for the experimental graph. Similarity was calculated to be 0.676 with a Matusita distance of 0.479 after running with the following parameters: random_seed = 42, g = 20, $\epsilon = 0.01$.

Reviewer #1:

c. Explore mechanistic insight via structure-based modeling

While the model performs impressively, the molecular mechanism underlying specificity remains abstract. I suggest that the authors select one representative protease-substrate pair (perhaps one with a well-characterized ΔH_{spec} mutation) and perform structural modeling (e.g., docking or AlphaFold3-based complex prediction). This could shed light on how specific residue-residue interactions enhance or weaken specificity, thereby linking statistical couplings to physical mechanisms.

Response 1.4:

We thank the Reviewer for their positive assessment of the performance of the ProSSpeC model. To further highlight the *interpretability* of ProSSpeC at single-residue resolution, we have broken down the top 50 residue couplings (in new Supplementary Fig. S6) that contribute most to the model prediction that EAPVp-mutSPV2cs is a functionally cleaving pair, whereas EAPVp-SPV2cs is not (which was experimentally demonstrated in Fig. 4c and new Fig. 5). As suggested by the Reviewer, we performed Alphafold3-based prediction of these two complexes to shed light on how statistical residue couplings might enhance or weaken specificity (new Fig. 4d).

- Of the top 10 residue couplings for EAPVp-mutSPV2cs, three amino acids in the Alphafold3-predicted EAPVp structure are <10 Å from the mutated P2-site (H->L): W206, N173, and F179.
- The top 1 coupling is a hydrophobic interaction between W206 and mutSPV2cs' P2 L. The wildtype P2 is H, a positively charged residue which may be unfavorable for EAPVp's W206.
- The remaining top 7 couplings are from EAPVp residues > 10 Å away from P2, which imply that cleavage specificity also arises from epistatic interactions within a distributed network of residues.

We have therefore added the following text in the Results section under the subsection "Engineering crosstalk for selective synoptosis":

"...To examine the basis of this prediction, we analyzed the top protease–substrate residue couplings contributing to the lower H_{spec} score. The ten strongest couplings all involved the mutated substrate P2 residue and numerous EAPVp residue positions, which were distributed across the protease (**Fig. 4d**, and **Supplementary Fig. S6**)."

And in the Discussion:

"To demonstrate the interpretability of the model, we analyzed a case where SPV2cs, which is not cleaved by EAPVp, was predicted to become cleavable upon a single amino acid substitution (**Fig. 4c**). We examined the top protease–substrate residue couplings contributing to this prediction. Of the ten strongest couplings, only three EAPVp residues were within 10 Å of the mutated substrate position (P2). The highest-ranked interaction was a hydrophobic contact between W206 and the mutant P2 Leu, whereas the wild-type P2 His, a positively charged residue, may be unfavorable for W206. The remaining seven top couplings involved residues located more than 10 Å from P2, suggesting that cleavage specificity arises partly from long-range epistatic interactions within a distributed residue network (**Fig. 4d** and **Supplementary Fig. S6**). These results illustrate how ProSSpeC captures interpretable residue–residue dependencies underlying substrate recognition."

Reviewer #1:

d. Discuss out-of-cleavage-site effects

In the Discussion, the authors should comment on the limitation of focusing solely on the cleavage region. Mutations outside the P6–P1' region (e.g., distal regions in the substrate protein) may alter local structure or dynamics, affecting cleavage. Acknowledging this limitation and suggesting future directions for extending the model (e.g., through structural context or experimental feedback loops) would strengthen the discussion.

Response 1.5:

We thank the Reviewer for this suggestion, we now added more information, new analysis and updated Fig. 4a, as well as commentary around the extended (20-a.a.) substrate context window in the Discussion:

"We demonstrate that ProSSpeC, a quantitative model for predicting protease-substrate interaction, enables the identification and engineering of protease-substrate pairs with enhanced cleavage activity in human cells. Unlike most previous analyses of Nla protease activity²³⁻²⁵, our ProSSpeC-guided experiments evaluated not only the 7-amino acid substrate but also its extended 20-amino acid context window, providing a more comprehensive assessment of protease-substrate specificity. Previous work by Beaumont, et al.²⁶ showed that extended substrate context matters for cleavage specificity. Our family-wide analysis reveals that residues flanking the cleavage motif play a role in cleavage efficiency. **Fig. 4a** shows that the composition of amino acids (and not just length) outside of the P6-P1' region significantly improves catalytic efficiency (1.3 to 7.9-fold). This provides evidence that the extended context contributes to proteolytic cleavage. One potential explanation may be that these regions have been subjected to evolutionary selection, potentially playing a role in the local structural dynamics of substrate recognition. Additionally, ProSSpeC can also predict cleavage activity at single-residue resolution (**Fig. 4b,c**), enabling programmable protease-substrate recognition. ..."

Reviewer #1:

e. Address substrate promiscuity

ProSSpeC is designed around specificity prediction, but some proteases may exhibit promiscuous behavior—cleaving multiple substrates with moderate affinity. Could the authors comment on how the model handles such cases? Does a moderately negative H_{spec} score reliably indicate partial activity across multiple substrates?

f. Clarify the CABMV case

The CABMV protease shows high cleavage efficiency toward its own substrate in the assay, but the H_{spec} score suggests potential promiscuity. This case deserves further discussion—does it suggest a decoupling between experimental signal strength and model specificity? Could this be a case of an energetically favorable but non-unique interaction? Understanding and explaining such discrepancies will help delineate the boundaries of the model.

Response 1.6:

In the expanded orthogonality matrix of 225 protease-substrate combinations (updated Fig. 3a,b), ProSSpeC H_{spec} scores correlated well with experimentally validated protease-substrate specificities (Pearson R = 0.68; DeltaCon similarity = 0.676). We can notice in the orthogonality matrix some off-diagonal regions that are signals of crosstalk. The model does a good job finding these regions where a single protease can interact with several peptides. To rigorously quantify this, we now generated a Receiver Operating Characteristic (ROC) curve for all 225 protease-substrate pairs in the orthogonality matrix. Using H_{spec} score as the predictor and assuming an experiment threshold for successful cleavage at the normalized fluorescence of 0.1, the area under the curve (AUC) = 0.87 and the optimal H_{spec} cutoff = -84 (new Supplementary Fig. S2 and Methods). When H_{spec} scores are far from the cutoff, ProSSpeC can reliably predict cleaving or noncleaving pairs. However, H_{spec} scores close to the cutoff are less reliable (most noticeably in CABMVp, EAPVp, and the newly added WMV2p and ACPp). Specifically, in these proteases, there are a lot of false positives being predicted. We noticed that there seem to be a protease sequence-dependent threshold, so we normalized H_{spec} score for each pair against the cognate H_{spec} score (i.e., scores along the diagonal), by dividing with cognate score (compare similarity between new Supplementary Table S6 and updated Fig. 3b). The AUC improved to 0.908, indicating that cognate-normalization improved discriminatory power (new Supplementary Fig. S2). The problem remains in that normalized H_{spec} scores close to the cutoff are still less

reliable. Nonetheless, normalization by cognate H_{spec} values reduces the number of false positives for these four proteases.

One possible explanation as to why there is a discrepancy between the predicted promiscuity and the experimentally high substrate specificity for these proteases is that they may contain attributes that deviate from shared Nla protease family properties. ProSSpeC is trained to detect specific signals across all Nla protease-substrate pairs, and while the model is reweighted to prevent bias from the training data, it is always possible that these Nla protease-substrate pairs are not well represented (i.e., rare) within the distributions found in natural sequence data. Expanding experimental coverage of low-homology sequences and retraining ProSSpeC accordingly could improve prediction accuracy and sequence diversity. This is briefly discussed in the Discussion, but we have expanded to include this commentary about CABMVp.

We thus now have added this statement in the Results section under the new second subsection "Experimental validation of the ProSSpeC model":

"... Also, using these pairs and to quantify how the model captures crosstalk in off-diagonal elements, we generated a Receiver Operating Characteristic (ROC) curve with an Area Under the Curve (AUC) = 0.878 showing that the model exhibits strong discriminatory ability between true cleaving and non-cleaving pairs (H_{spec} cutoff = -84)."

Also, these moderately negative H_{spec} scores may denote that protease-substrate pairs do bind but do not result in actual cleavage. We have added new text in Discussion describing this feature of the model and how we can explain some observed crosstalk as well as correct some false positives. The new text reads:

"One limitation shared by predictive models, including ProSSpeC, is the lower accuracy for sequences (or pairs of sequences) that significantly deviate from the natural distribution. For example, ProSSpeC typically predicts low specificity for substrates with non-glutamine amino acids at the highly conserved P1 site. Another example is the predicted promiscuity but experimentally high substrate specificity for CABMVp in **Fig. 3**. These results suggest that CABMVp may contain attributes that deviate from the learned Nla protease family properties. Normalizing noncognate H_{spec} scores relative to the cognate H_{spec} score enhanced discriminatory performance and decreased the incidence of false-positive predictions for CABMVp and other proteases. (**Supplementary Fig. S2** and **Supplementary Table S6**). Still, to properly address this limitation, the low-homology sequence space needs to be experimentally explored. Retraining ProSSpeC on an expanded dataset could enhance its predictive power and expand the sequence diversity of engineered protease-substrate pairs. Nonetheless, we have shown that the current model is effective at engineering pairs for an expanded distribution based on natural sequence properties, supporting immediate use and application of ProSSpeC."

Reviewer #1 (Remarks on code availability):

I failed to access the code as it shows "this connection is not private".

Response 1.7:

The website's security certificate was temporarily expired. It has now been fixed, and a regularly scheduled job was added to the server to ensure that the security certificate is always valid.

Reviewer #2 (Remarks to the Author):

Reviewer #2 (Remarks on code availability):

Clear and easy to use

Response 2.1:

We thank Reviewer #2 for their positive feedback.

Reviewer #3 (Remarks to the Author):

In the manuscript "Identification and engineering of highly functional synthetic proteases in cells using co-evolutionary models," the authors present Protease Substrate Specificity Calculator (ProSSpeC), a co-evolution-guided computational tool that predicts protease-substrate specificity. Using direct coupling analysis, ProSSpeC quantifies protease-substrate specificity via a Hamiltonian score. ProSSpeC was trained on over 3,000 natural Nla protease-substrate pairs from the Potyviridae family. It accurately predicted cleavage efficiency for many protease-substrate pairs, with Hamiltonian scores strongly correlated with experimental cleavage-induced fluorescence in human cells. Several proteases were found to outperform the commonly used TEV protease (TEVp) in human cells. Additionally, ProSSpeC predicted the effects of single-point mutations in the cleavage sites of substrates, enabling tuning of substrate specificity. The authors further used ProSSpeC to reprogram protease specificity and engineer crosstalk between a protease and a mutant non-cognate substrate. As a demonstration, they used a protease to selectively trigger apoptosis in cells expressing engineered caspase-3 containing the mutant cleavage site of another protease.

Overall, this manuscript presents a co-evolutionary computational model supported by cell-based experimental validation. The model quantitatively predicts protease-substrate specificity at single-residue resolution. Integration of modeling with cell-based assays establishes a 'predict-validate' story. This study can offer programmable protease tools in synthetic biology, potentially enabling therapeutic applications such as selective cell killing.

Response 3.1:

We thank the reviewer for taking the time to review our work. We are indeed excited to share the programmable proteases to the broader synthetic biology community. We hope we have sufficiently addressed your specific concerns in this new revision, as discussed point-by-point below.

Reviewer #3

Major Comments:

In the cellular fluorescence assay, it would be helpful if both the protease and the substrate were tagged with different fluorescent markers. That way, expression levels of both components can be monitored and controlled for by flow cytometry gating. Without rigorous quantification, it is difficult to tell whether differences in activity are really due to actual proteolysis efficiency or just variable expression or stability. Some data on protease and substrate levels would be necessary.

Response 3.2:

We have now done substantial, additional experiments wherein each protease (and reporter) was tagged with sfGFP and each substrate was tagged with mTagBFP2. New Supplementary Fig. S11 shows the correlation between cleavage-induced fluorescence quantified through fluorescence microscopy imaging (Method 1: without fluorescent protein tags for either protease or substrate) and cleavage-induced fluorescence quantified through flow cytometry (Method 2: fluorescent protein-tagged protease and substrate). We observed strong correlations between both Methods, whether cleavage-induced fluorescence was normalized to protease amount (Pearson $r = 0.80$), to substrate amount (Pearson $r = 0.82$), or to both (Pearson $r = 0.81$). We have now added these in the Methods section under "Cleavage-induced fluorescence assay":

"To benchmark the accuracy of the fluorescent protein reconstitution assay, the 31 pairs were also tested where each protease (and the nuclear reporter) was tagged to sfGFP and each substrate to mTagBFP2. Cells were then analyzed through flow cytometry (BD LSRFortessa™ Cell Analyzer). Cleavage-induced fluorescence intensities normalized against protease amount, substrate amount,

or both yielded strong correlations (Pearson $r = 0.8$ ($p = 1.9 \times 10^{-26}$), 0.82 ($p = 1.5 \times 10^{-28}$), and 0.81 ($p = 7.8 \times 10^{-27}$), respectively) with the previous assay, suggesting that both assays reflect the actual efficiency of proteolysis (**Supplementary Fig. S11**)."

Reviewer #3

In the "ProSSpeC-guided engineering of nonnative cleavage specificity" section, it would add value to the paper if the authors systematically profile a library of substrate mutations for TEVp and TVMVp, which are widely used. Knowing how to improve cleavage efficiency for these proteases through targeted mutations would be valuable for many synthetic biology researchers. Right now, the focus is on ScMVp, SbMVp, and EAPVp, which are less common and could limit potential protease-based applications.

Response 3.3:

We thank the reviewer for this suggestion. We now have tested 12 additional triple mutants for TEVp, TVMVp, as well as the lesser known LMVp, TVBMVp, PMAVp, and WMV2p (with similar or better H_{spec} against their wild-type substrates). Supplementary Fig. S3 show that these mutants have essentially similar cleavage efficiencies compared to their wildtype counterparts. We note that mutations can easily destroy protein function, but here we show that ProSSpeC-guided mutations efficiently maintain the enzyme-substrate specificity relationship, allowing one to explore sequence space using the H_{spec} score as a prediction of functionality. The Hamiltonian score is able to quantify from a larger potential sequence space: those mutations that maintain amino acid couplings even if the sequence is different. Evolutionarily, this represents a space of neutral or nearly-neutral mutations that is required to provide robustness and maintain functionality even when mutations occur. We have added the following statement under the "ProSSpeC-guided engineering of nonnative cleavage specificity" section of the main text:

"To determine whether ProSSpeC can be used to predict protease mutations that improve or preserve catalytic efficiency, we mutated five proteases, including the widely used TEVp and TVMVp, with similar H_{spec} scores to their wildtype counterparts. We found that these mutants have comparable efficiencies to wildtype (**Supplementary Fig. S3**), demonstrating the ability of ProSSpeC to explore sequence space using H_{spec} as a proxy for functionality."

Reviewer #3:

ProSSpeC offers quantitative Hamiltonian scores, but does not provide mechanistic insight or structural rationale for the most favorable protease-substrate pairs. Consider using AlphaFold-predicted complex structures or contact maps to rationalize top-scoring interactions.

Response 3.4:

Similar to **Response 1.4** and as also suggested by this Reviewer, we performed AlphaFold3-based prediction of these two complexes to shed light on how statistical residue couplings might enhance or weaken specificity (new Fig. 4d).

- Of the top 10 residue couplings for EAPVp-mutSPV2cs, three amino acids in the AlphaFold3-predicted EAPVp structure are $< 10 \text{ \AA}$ from the mutated P2-site (H->L): W206, N173, and F179.
- The top 1 coupling is a hydrophobic interaction between W206 and mutSPV2cs' P2 L. The wildtype P2 is H, a positively charged residue which may be unfavorable for EAPVp's W206.
- The remaining top 7 couplings are from EAPVp residues $> 10 \text{ \AA}$ away from P2, which imply that cleavage specificity also arises from epistatic interactions within a distributed network of residues.

We have therefore added the following text in the Results section under the subsection “Engineering crosstalk for selective synoptosis”:

“...To examine the basis of this prediction, we analyzed the top protease–substrate residue couplings contributing to the lower H_{spec} score. The ten strongest couplings all involved the mutated substrate P2 residue but different EAPVp positions, which were distributed across the protease (**Fig. 4d**, and **Supplementary Fig. S6**).”

And in the Discussion:

“To demonstrate the interpretability of the model, we analyzed a case where SPV2cs, which is not cleaved by EAPVp, was predicted to become cleavable upon a single amino acid substitution (**Fig. 4c**). We examined the top protease–substrate residue couplings contributing to this prediction. Of the ten strongest couplings, only three EAPVp residues were within 10 Å of the mutated substrate position (P2). The highest-ranked interaction was a hydrophobic contact between W206 and the mutant P2 Leu, whereas the wild-type P2 His, a positively charged residue, may be unfavorable for W206. The remaining seven top couplings involved residues located more than 10 Å from P2, suggesting that cleavage specificity arises partly from long-range epistatic interactions within a distributed residue network (**Fig. 4d** and **Supplementary Fig. S6**). These results illustrate how ProSSpeC captures interpretable residue–residue dependencies underlying substrate recognition.”

Reviewer #3:

The manuscript does not compare ProSSpeC against alternative protease-substrate specificity predictors, such as PROSPER, PeptideCutter, and iProt-Sub. A side-by-side comparison, even just for a subset of known pairs, would strengthen the claim that coevolutionary modeling adds value.

Response 3.5:

We thank the Reviewer for this thoughtful comment and recognize that several computational models exist for general protease cleavage, however we found them to not be applicable to our specific evaluation of N1a proteases:

1. PeptideCutter only evaluates the Tobacco Etch Virus protease (TEVp) from the N1a peptidase family and does not give any indication of specificity, other than TEVp will cut at the glutamine residue if present at P1 and not otherwise. Mutating one residue in the 7-amino acid substrate motif sequence has no effect on this prediction, mutating more than one amino acid automatically changes the prediction to no cutting. Furthermore, mutations in the extended context have no impact on these TEVp predictions.
2. ProsperousPlus (now the location of PROSPER and iProtSub) also hosts many protease predictions, but it does not support the N1a protease family, including key enzymes like TEVp.

Nonetheless, we do recognize the work of other scientists in this field, so we have chosen to highlight other useful resources across protease families in our Discussion:

“While other models for protease cleavage exist, they focus on protease families other than N1a proteases²⁷⁻³⁰, and more recently, only on TEVp²⁵. By contrast, our ProSSpeC model provides full support for the entire N1a protease family (including TEVp) and leverages coevolutionary information to guide N1a protease-substrate specificity predictions, allowing for a comprehensive analysis of the underlying sequence space of both enzyme and substrate. We believe that our tool

fills an important research niche that complements existing resources, providing computational support that did not previously exist.”

Reviewer #3:

The observation that extending substrate sequences from 7 to 20 amino acids improves cleavage is interesting, but it is only shown for a few cases. Also, it would be good to clarify what the additional 13 amino acids are. Are they from natural sequence context, or are they synthetic linkers?

Response 3.6:

The amino acids for the extended context (20 amino acids) comes from the natural sequence of the Potyviral polyprotein that surrounds the 7-amino acid motif. Supplementary Table S7 contains all the extended substrate sequences for all experiments. We have updated the legend in updated Fig. 4a to clarify the exact substrate sequences used in the different test conditions.

Reviewer #3:

Could flexible sequences like GGGs repeats also boost cleavage? Would the position of the 13 amino acids matter, such as on the N-terminus or the C-terminus of the 7-amino acid cleavage site?

Response 3.7:

We thank the reviewer for this comment. We have now added new data on protease activity against GS linker-flanked cleavage motifs in updated Fig. 4a. While there are cases where this shows improvement over the 3x7 amino acid cleavage motifs, most cases show that the extended motif (20 amino acid natural cleavage sequence) does improve cleavage efficiency. We have added the following statement in the Results section under the new second subsection “Experimental validation of the ProSSpeC model”:

“...Guided by the H_{spec} score (**Supplementary Table S4**), substituting the triplet repeat of the 7-amino-acid substrate with the corresponding 20-amino-acid natural sequence resulted in a 1.3- to 7.9-fold increase in cleavage-induced fluorescence (**Fig. 4a**). While GS linkers flanking the 7-amino-acid motif enhanced cleavage in some cases, the 20-amino-acid natural context consistently produced higher activity....”

Reviewer #3:

The fluorescence signal for some protease-substrate pairs seems to plateau, potentially masking finer distinctions. It would be helpful to report absolute fluorescence ranges and dynamic range limits for the assay. Consider using titration experiments to better quantify proteolytic efficiency.

Response 3.8:

Supplementary Fig. S1 shows quite a wide dynamic range such that the *relative* efficiencies are sufficiently captured by the assay. In new Supplementary Fig. S11, we now present additional cleavage data on all 31 proteases and show that the cleavage-induced fluorescence values are well below the maximum possible sfCherry/sfGFP achieved by the positive control condition. In new Supplementary Fig. S12, we have also done dose titration on 5 representative proteases, which show that the protease amount we used in all other experiments (1X = 16.6 ng) is indeed at saturation, indicating that the cleavage-induced fluorescence values reflect their relative activities. We have added this to the Methods section under the subsection “Protease titration”:

"Using the same transfection and analysis protocol, we tested five proteases across a range of plasmid concentrations (1.66 ng, 5 ng, 16.6 ng, and 50 ng) to evaluate their proteolytic efficiency. As protease expression increased, some proteases showed higher activity while others reached an early plateau (**Supplementary Fig. S12**). Notably, all proteases reached fluorescence saturation at 16.6 ng, the concentration used throughout this study. The resulting fluorescence therefore reflects apparent catalytic efficiencies that can be directly compared across proteases."

Reviewer #3:

The ProSSpeC model is validated using a relatively limited number of 10×10 protease-substrate pairs. It would be great to see more pairs tested to better assess how generalizable the model is.

Response 3.9:

We thank the reviewer for this comment. We have more than double the pairs tested from 100 to 225 protease-substrate pairs, giving us a 15x15 orthogonality matrix (in updated Fig. 3).

Reviewer #3:

Minor Comments:

Can the authors comment on how this tool can or cannot be applied beyond the Potyviridae family? Would the same modeling approach generalize to other protease families?

Response 3.10:

This tool only applies to the *Potyviridae* family. The method can certainly be applied to other protease families if retrained on sequence datasets for other families. The ProSSpeC model itself cannot generalize across protease families without retraining.

Reviewer #3:

It would be helpful to indicate where TEV protease lands on the distribution in Fig. 2a—TEVp is a standard point of comparison, so readers will naturally look for it.

Response 3.11:

We have updated Fig. 2a to indicate where TEVp is.

Reviewer #3:

Many experiments used only two independent replicates. N = 3 or more would be ideal.

Response 3.12:

All experiments are now $N \geq 3$.

Reviewer #3:

In Fig. 4b, it is not entirely clear what each dot represents. Clarifying that would help readers interpret the plot.

Response 3.13:

We have clarified what each dot represents in the caption for Fig. 4b:

"b, DeltaH (ΔH_{spec}) versus DeltaFluor (Δ fluorescence) for each dot, which represents the difference between a protease cleaving one substrate versus the same protease cleaving another substrate (see also Supplementary Table S5)."

Reviewer #3:

Fig. 4d could benefit from a few more control conditions where the cell expresses "EAPVp + Caspase3-EAPVcs" and "SPV2p + Caspase3-SPV2cs".

Response 3.14:

We have placed all synoptosis experiment results in new Fig. 5 including positive controls "EAPVp + Caspase3-EAPVcs" and "SPV2p + Caspase3-SPV2cs".

Reviewer #3:

Define synoptosis more clearly on first mention.

Response 3.15:

We now have clarified the definition of synoptosis under the subsection "Engineering crosstalk for selective synoptosis":

"...We tested whether EAPVp can synthetically trigger caspase-mediated apoptosis (i.e., synoptosis)..."

Reviewer #3:

Lastly, the authors wrote: "Our results highlight the potential of engineered proteases to interface with endogenous cellular pathways." However, there is no demonstration of such interfacing in this study.

Response 3.16:

We apologize for the confusion. We now added a statement clarifying that the engineered Caspase3 used in the study interfaces with the endogenous cellular pathway that result in apoptosis (2nd paragraph of "Engineering crosstalk for selective synoptosis" subsection:

"...To demonstrate this potential, we tested whether proteases can be used to selectively kill cells^{21,22} that harbor a point mutation in a target protein. We used and engineered Caspase3, an executioner caspase whose activation leads to cleavage of numerous endogenous proteins, ultimately leading to apoptosis."

Reviewer #3 (Remarks on code availability):

n/a

Reviewer #4 (Remarks to the Author):

Here the authors developed ProSSpeC, a generative model that was trained using 3817 pairs of viral Nuclear Inclusion a (Nla) proteases and their cleavage peptide sequences. The model can generate a score (H_{spec}) for an input, with lower scores (negative scores) predicts higher chance of cleavage. The authors then validated this model using a cell-based assay based on reconstitution of fluorescence protein (successful cleavage releases a fragment, which enters the nucleus and complements the other fragment in the nucleus to form a complete fluorescent protein).

The writing is exceedingly short. This is not acceptable. The whole paper reads like a summary, instead of a full-length scientific paper. Lack of details renders it difficult to review this paper as many Figures and authors' interpretations are difficult to understand.

Response 4.1:

We have now included additional details to better explain concepts and Figures in the paper. Details regarding the H_{spec} model were added to the Results section, along with the rationale of its significance. We hope that the following changes provide readers with a stronger intuition of the model as they read through the Results:

“ProSSpeC is a coevolutionary model that leverages the covariation of Nla protease sequences along with their aligned substrate sequences to calculate a Hamiltonian specificity score (H_{spec}), where the more negative the H_{spec} score is, the stronger the predicted specificity is. This specificity score is defined by the Potts Hamiltonian energy of canonical protease-substrate pairs (Equation 1), with the Potts Hamiltonian energy of scrambled enzyme-substrate pairs (Equation 2) removed, to provide a score of sequence attributes unique to interacting pairs and not shared due to family-likeness or general sequence similarity (Equation 3). For both canonical and scrambled protease-substrate pairs, the Potts Hamiltonian is defined as the sum of all parameters in a Boltzmann distribution, with couplings (e_{ij}) and local fields (h_i) inferred using direct coupling analysis (DCA), specifically the mean-field formulation (mfDCA)⁹. A similar rationale for using the DCA Hamiltonian to characterize molecular interactions has been applied to study specificity in two-component systems¹², recognition in protein-RNA interactions¹⁷, and to predict and engineer compatibility in hybrid transcription factors^{13,18}. The ProSSpeC model applies this approach to quantify interactions of the protease and the 7-amino acid substrate; we masked out substrate residues outside of this window in H_{spec} calculations to yield the masked H_{spec} (Equation 4). The masked H_{spec} was utilized for predictions where the designed experiment did not include the extended substrate context. ProSSpeC allows us to quantitatively assess the specificity of Nla proteases and their substrate motif through sequence alone, allowing us to filter and rank both natural Nla protease-substrate pairs as well as mutated ones prior to experimental testing.”

Reviewer #4:

The study is limited to Nla protease family. No other types of proteases are examined. Authors need to tune down all broad claims of “proteases” and spell out clearly that the work is limited to Nla proteases in the title and abstract.

Response 4.2:

We have gone through the text and specified that we are looking at Nla protease-substrate interactions in several places where clarification was needed.

Reviewer #4:

There is no "engineer protease" in this manuscript. Authors only tested mutated substrate sequences. The last paragraph of the "Discussion" needs to be corrected.

Response 4.3:

We thank the reviewer for this comment and recognize that we needed to highlight more information on the proteases themselves. When we performed *in silico* mutation of the proteases, we found very few mutants that had H_{spec} scores that are similar to, if not better than, their wild-type counterparts. Of those that we did find and test, they still maintained similar functionality to their wild-type counterparts. This is now shown in new Supplementary Fig. S3, where five proteases were mutated at three sites, and we saw little to no difference in cleavage-induced fluorescence. This work will be expanded in the future, where we will attempt to mutate both enzyme and substrate in conjunction with each other to access new sequence spaces. Thus, we have now added this paragraph in the Results section under the subsection "ProSSpeC-guided engineering of nonnative cleavage specificity":

*"To determine whether ProSSpeC can be used to predict protease mutations that improve or preserve catalytic efficiency, we mutated five proteases, including the widely used TEVp and TVMVp, with similar H_{spec} scores to their wildtype counterparts. We found that these mutants have comparable efficiencies to wildtype (**Supplementary Fig. S3**), demonstrating the ability of ProSSpeC to explore sequence space using H_{spec} as a proxy for functionality."*

Reviewer #4:

It is very strange that Fig. 1-4 are all covered by the first sub-title of the "Result" section, and the main Figures only have these 4 Figures. Many supplementary Figures should be moved to main Figures.

Response 4.4:

We have broken up Fig. 4 to create new Figs. 4 and 5. In new Fig. 4d, we included AlphaFold3 models that attempt to explain why the engineered EAPVp-mutSPVcs pair works. In new Fig. 5, we also added additional experimental protease-substrate combinations to comprehensively showcase the synoptosis use case, as well as to improve the flow of the text.

Reviewer #4:

There are many cases that experimental data do not fit their Hspec prediction. Authors need to explain all these pairs that do not fit their models: Fig. 2b: SPV2p? PTVp? Fig. 3b: PPV versus KoMV substrate? Fig. S2: TEV: the score of -249 showed higher signal than -296. For PeMoV, -243 showed lower signal than -231? Fig. S3: for ScMV: the blue (-184) was predicted to be stronger than orange (-153), but the sfCherry3C signaling is higher for orange, how do you explain this? Fig. 4a: increase length to 20 did not drastically increase the degree of cleavage for ISMVp etc.

Response 4.5:

While the model is not perfect, H_{spec} scores and experimental fluorescence values are shown to be correlated (Fig. 3, Pearson R = 0.68; Fig. 4b, Pearson R = -0.67). Discrepancies between predicted and experimental values may be attributed to the model not being able to delineate between sequence effects on substrate binding vs actual substrate cleavage (e.g., a more negative H_{spec} score may indicate higher binding affinity but not necessarily increased cleavage efficiency). Also, small differences in ProSSpeC scores

are less reliable than large differences. Additionally, due to the sequence-based nature of the model, Nla enzyme-substrate pairs which do not adhere to the natural distributions found across the family are less reliable. This is addressed in the Discussion and is already a point of interest for future work, where we can improve our modeling capabilities by expanding sequence pair novelty in the training data.

“One limitation shared by predictive models, including ProSSpeC, is the lower accuracy for sequences (or pairs of sequences) that significantly deviate from the natural distribution. For example, ProSSpeC typically predicts low specificity for substrates with non-glutamine amino acids at the highly conserved P1 site. Another example is the predicted promiscuity but experimentally high substrate specificity for CABMVp in **Fig. 3**. These results suggest that CABMVp may contain attributes that deviate from the learned Nla protease family properties. Normalizing noncognate H_{spec} scores relative to the cognate H_{spec} score enhanced discriminatory performance and decreased the incidence of false-positive predictions for CABMVp and other proteases. (**Supplementary Fig. S2** and **Supplementary Table S6**). Still, to properly address this limitation, the low-homology sequence space needs to be experimentally explored. Retraining ProSSpeC on an expanded dataset could enhance its predictive power and expand the sequence diversity of engineered protease-substrate pairs. Nonetheless, we have shown that the current model is effective at engineering pairs for an expanded distribution based on natural sequence properties, supporting immediate use and application of ProSSpeC.”

We now also generated a Receiver Operating Characteristic (ROC) curve for all 225 protease-substrate pairs tested. Using H_{spec} score as the predictor and assuming an experiment threshold for successful cleavage at the normalized fluorescence of 0.1, the area under the curve (AUC) = 0.87 and the optimal H_{spec} cutoff = -84 (new Supplementary Fig. S2 and Methods). We thus now have added this statement in the Results section under the new second subsection “Experimental validation of the ProSSpeC model”:

“... Also, using these pairs and to quantify how the model captures crosstalk in off-diagonal elements, we generated a Receiver Operating Characteristic (ROC) curve with an Area Under the Curve (AUC) = 0.878 showing that the model exhibits strong discriminatory ability between true cleaving and non-cleaving pairs (H_{spec} cutoff = -84).”

Reviewer #4:

Fig. 3a: aren't these pairs used to train the model? Why it is a surprise that they correlate?

Response 4.6:

ProSSpeC is an unsupervised model trained on natural sequences. The correlation is between the ProSSpeC's predicted score and the experimental functional cleavage assay. Beyond the cognate protease-substrate pairs, the unsupervised model is also able to capture (off-diagonal) crosstalk interactions.

Reviewer #4:

The meaning of Fig. 4d is not clear: there is no engineering of protease. The authors just put a mutant substrate peptide sequence into caspase3. In this case, wouldn't it be better to use the native substrate of this protease (EAPVp)?

Response 4.7:

We apologize for any confusion in Fig. 4d (now new Fig. 5) and have added to the text that Caspase3 is the protease, which we have engineered in this specific plot. This figure conceptualizes whether or not point mutations in a target protein can be detected by a designer protease in a mixed population of cells. The experiments represent a proof-of-concept for our ultimate goal of creating mutation-detecting proteases. We hope to leverage the findings from this work to expand both protease sequences and substrate sequences into a library of targetable sequence motifs and engineered proteases, while also understanding off-target effects. Here, we were able to leverage off-target effects for synoptosis.

REVIEWERS' COMMENTS

Reviewer #1 (Remarks to the Author):

The authors have addressed all my critiques in the revised manuscript. I am supportive of publication.

Reviewer #3 (Remarks to the Author):

The revision addresses all of our concerns.

RESPONSE TO REVIEWERS:

We thank the Reviewers for their positive feedback on the revised manuscript.